



# Cold vs. warm water route — sources for the upper limb of the AMOC revisited in a high-resolution ocean model

Siren Rühs[1], Franziska U. Schwarzkopf[1], Sabrina Speich[2], and Arne Biastoch[1]

[1]GEOMAR Helmholtz Centre for Ocean Research Kiel, Kiel, Germany
[2]LMD-IPSL, UMR 8539, Département de Géosciences, ENS, PSL Research University, Paris, France

**Correspondence:** Siren Rühs (sruehs@geomar.de)

**Abstract.** The northward flow of the upper limb of the Atlantic Meridional Overturning Circulation (AMOC) is fed by waters entering the South Atlantic from the Indian Ocean mainly via the Agulhas Current (AC) system and by waters entering from the Pacific through Drake Passage (DP), commonly referred to as the 'warm' and 'cold' water routes, respectively. However, there is no final consensus on the relative importance of these two routes for the upper limb's volume transport and thermohaline properties. In this study we revisited the AC and DP contributions by performing Lagrangian analyzes between the two source regions and the North Brazil Current (NBC) at 6°S in a realistically forced high-resolution (1/20°) ocean model.

Our results agree with the prevailing conception that the AC contribution is the major source for the upper limb transport of the AMOC. However, they also suggest a non-negligible DP contribution of at least 40 %, which is substantially higher than estimates from previous Lagrangian studies with coarser resolution models, but now better matches estimates from Lagrangian observations. Moreover, idealized analyzes of decadal changes in the DP and AC contributions indicate that the ongoing increase in Agulhas leakage indeed may have evoked an increase in the AC contribution to the upper limb of the AMOC while the DP contribution decreased. In terms of thermohaline properties, our study highlights that the AC and DP contributions cannot be unambiguously distinguished by their temperature, as the commonly adopted terminology may imply, but rather by their salinity when entering the South Atlantic. During their transit towards the NBC the bulk of DP waters experiences a net density loss through a net warming, whereas the bulk of AC waters experiences a slight net density gain through a net increase in salinity. Notably, these density changes are nearly completely captured by those Lagrangian particle trajectories that reach the surface mixed layer at least once during their transit, which amount to 66 % and 49 % for DP and AC waters, respectively. This implies that more than half of the water masses supplying the upper limb of the AMOC are actually formed within the South Atlantic, and do not get their characteristic properties in the Pacific and Indian Oceans.





# 1 Introduction

Within the framework of the global overturning circulation, the South Atlantic is depicted as a conduit, exporting cold dense waters produced in the North Atlantic into the Indian and Pacific basins, and enabling the return flow of warmer lighter waters back into the North Atlantic which in turn feed the deep water formation (Broecker, 1991; Richardson, 2008). The resulting

net equatorward heat transport within the South Atlantic and thus northward heat transport across the whole Atlantic (cf. Kelly et al., 2014) is unique among all ocean basins and has been long recognized for its importance in modulating European (e.g., Palter, 2015; Moffa-Sánchez and Hall, 2017) as well as global (e.g., Srokosz et al., 2012; Buckley and Marshall, 2016; Lynch-Stieglitz, 2017) climate. Even though variability in the overall overturning strength and associated heat transport have been mostly related to variability in deep water formation (e.g., Biastoch et al., 2008a; Yeager and Danabasoglu, 2014), the involved

convection process is highly sensitive to the local stratification and thus has been suggested to be additionally impacted by the thermohaline properties of the northward upper layer return flow (Haarsma et al., 2011; Garzoli and Matano, 2011; Cimatoribus et al., 2012; Garzoli et al., 2013).

The northward upper layer return flow, that is, the upper limb of the meridional overturning circulation in the Atlantic (AMOC), is mainly supplied by waters entering the South Atlantic south of America through Drake Passage (Pacific-Atlantic

route) and south of Africa via the Agulhas Current System (Indo-Atlantic route). The Pacific-Atlantic route is characterized by cold and fresh waters (Rintoul, 1991) flowing from the Antarctic Circumpolar Current (ACC) directly into the South Atlantic subtropical gyre without any intermediate recirculation in other basins. Within the literature it has been also referred to as the direct 'cold water route' (Speich et al., 2001). The Indo-Atlantic route injects comparatively warm and salty waters into the South Atlantic (Gordon, 1986; Richardson, 2007) in form of large anticyclonic eddies (Agulhas rings), smaller cyclonic

eddies, and filaments shed at the retroflection of the Agulhas Current, comprehensively termed Agulhas leakage (de Ruijter et al., 1999; Lutjeharms, 2006). It has been commonly referred to as the 'warm-water route' (Gordon, 1986; Speich et al., 2001).

Due to their remarkably different water properties the ratio of the inflow through the Pacific-Atlantic and Indo-Atlantic routes impacts the thermohaline characteristics of South Atlantic water masses — yet, there is no final consensus on the

relative importance of the two routes for the thermohaline properties and northward transport of the upper limb of the AMOC (Garzoli and Matano, 2011; Dong et al., 2011). Gordon (1986) postulated the Indo-Atlantic route to be the major source for the upper limb of the AMOC (warm water route hypothesis), while Rintoul (1991) found only a minor importance of the Indo-Atlantic route and suggested the Pacific-Atlantic route to be the dominant one (cold water route hypothesis). Since then, both hypotheses have been controversially discussed. For instance, Schmitz (1995), Macdonald (1998), and Sloyan and Rintoul

(2001) supported the cold water route hypothesis, whereas Speich et al. (2001, 2007), Holfort and Siedler (2001), Donners and Drijfhout (2004), Rodrigues et al. (2010), and Dong et al. (2011) favored the warm water route hypothesis. Moreover, Speich et al. (2001, 2002, 2007) suggested that the warm and cold water route debate is more complex than previously stated due to the impact of the southern hemisphere 'supergyre' spanning all three basins. They show that the warm water route, which was traditionally assumed to consist of Indian Ocean waters originating from the Indonesian Throughflow (Gordon, 1986; Le





Bars et al., 2013), also contains waters entering the Indian Ocean through Tasman leakage south of Australia (Speich et al., 2001, 2002), and waters originating indirectly from Drake Passage through a connection via the Antarctic Circumpolar Current (ACC) and subsequent circulation in the Indian Ocean part of the supergyre (note that some studies also referred to the indirect Drake Passage contribution as the 'indirect cold water route'). The relative volumetric contributions of the different sources to
the Indo-Atlantic route, as well as their pathways through the Indian Oceans and associated along-track property modifications have been recently comprehensively discussed in Durgadoo et al. (2017).

The conflicting views on the relative importance of the Indo-Atlantic and Pacific-Atlantic routes for the thermohaline properties and northward transport of the upper limb of the AMOC have been partially attributed to conceptual (such as diverting definitions of the cold and warm water routes) and methodological differences (such as the usage of Eulerian vs. Lagrangian
approaches to estimate the interbasin exchanges). Most importantly, Donners and Drijfhout (2004) refuted the majority of studies in support of the cold water hypothesis by arguing that their common choice of using inverse box model calculations based on distinct hydrographic sections is generally not suited to estimate the highly variable and intermittent interbasin exchange South of Africa via Agulhas Leakage. Since then, Lagrangian (model) analyzes have become the preferential tool to estimate Agulhas leakage (e.g., Richardson, 2007; Biastoch et al., 2008c; van Sebille et al., 2010; Durgadoo et al., 2013), and the warm
water route hypothesis has been prevailing. Still, the studies supporting it were mostly based on the evaluation of relatively coarse resolution non-eddying or eddy-permitting ocean model simulations (Speich et al., 2001; Donners and Drijfhout, 2004; Speich et al., 2007), and various studies (e.g., Biastoch et al., 2008c; Durgadoo et al., 2013) have demonstrated that the obtained estimates for the interocean exchange south of Africa are highly resolution dependent. They specifically pointed out that coarser non-eddying ocean models overestimate the strength of Agulhas leakage, thus necessitating a confirmation or revision
of the warm water route hypothesis by means of Lagrangian analysis with higher resolution eddy-rich flow fields.

Moreover, despite a general agreement that the knowledge on the characteristic thermohaline properties of the waters supplying the upper limb of the AMOC and their potential modification during their transit through the South Atlantic is of fundamental importance — including the individual roles played by surface, central, and intermediate waters — respective thorough analyzes within the Lagrangian framework have been scarce (Speich et al., 2001, 2007; Rimaud et al., 2012).

Finally, over the last decades, research on (potential) changes in the Agulhas region have become prominent, since (i) it has been shown that changes in the Agulhas region have the potential to influence the strength of the AMOC through different processes on various timescales (Weijer et al., 2002; Knorr and Lohmann, 2003; Biastoch et al., 2008b; Beal et al., 2011); and (ii) simulations with ocean and coupled ocean-atmosphere models indicate a strengthening of Agulhas leakage since the 1960s (Biastoch et al., 2009; Cheng, 2018). A conclusive observational corroboration of this long-term increase, however, is
still lacking (Backeberg et al., 2012; Le Bars et al., 2014), which may be due to decadal leakage variability related to the Southern Annular Mode (Biastoch et al., 2009) that tend to mask its long-term trend. The increase in Agulhas leakage has been attributed to an increase in the Southern Hemisphere westerlies (Durgadoo et al., 2013; Biastoch et al., 2015) and is expected to continue under global warming conditions (Biastoch and Böning, 2013). A potential related increase of salt input into the South Atlantic and its northward advection could oppose the anticipated weakening of the AMOC due to increasing freshwater
input in the North Atlantic (Beal et al., 2011). These notional causalities motivated several Lagrangian studies on the pathways



and associated advective timescales of Agulhas waters into the North Atlantic (van Sebille et al., 2011; Rühs et al., 2013). However, none of these studies directly investigated potential changes of the advective pathways, thermohaline properties and transports of the upper limb of the AMOC associated with changes in Agulhas Leakage, or tried to also relate those to changes in the Pacific-Atlantic contribution.

In this study we address the gaps outlined above and revisit the relative importance of the Indo-Atlantic and Pacific-Atlantic routes for the AMOCs upper limb volume transport and water mass characteristics by means of Lagrangian connectivity analyzes between the North Brazil Current (NBC) at 6°S and the two source regions in a realistic set-up of a high-resolution (1/20°) ocean model. In addition, for the first time, idealized analyzes of decadal changes in the individual contributions are presented. The NBC is chosen as reference region, since it merges all upper and intermediate northward flow of the tropical

South Atlantic and thus channels the upper limb of the AMOC in the tropics. It has been further shown that its transport variability captures decadal AMOC changes and may be used as an index for the AMOC strength (Rühs et al., 2015). Thus, choosing the NBC as reference region also allows for the investigation of potential decadal changes in the ratio of the two contributions.

## 2 Materials and methods

The bases for this study are offline Lagrangian analyzes of simulated three-dimensional time-varying eddy-rich velocity fields. The underlying model simulation and its performance are described in Sect. 2.1, whereas in Sect. 2.2 the Lagrangian experiment set-up is specified.

### 2.1 Hindcast simulation with eddy–rich ocean model configuration INALT20

The analyzed model output stems from a hindcast experiment (1958–2009) performed with the global nested eddy-permitting

to eddy-rich ocean/sea-ice model configuration INALT20 (Schwarzkopf et al., in preparation), which is a successor of the well-established INALT01 (Durgadoo et al., 2013) with higher resolution and a southward extended nest.

INALT20 was developed within the DRAKKAR framework (Barnier et al., 2014) and consists of an ocean model formulated with the Nucleus for European Modelling of the Ocean (NEMO, version 3.6, Madec and NEMO-team, 2016) coupled to the LIM2-VP sea-ice model (Fichefet and Maqueda, 1997). The ocean is therein described by the primitive equations, that are, the

Navier-Stokes equations along with a nonlinear equation of state, with commonly adopted approximations resulting from scale considerations including the Boussinesq approximation that effectively reduces mass conservation to volume conservation. It is implemented on a horizontal tripolar Arakawa C-grid (Mesinger and Arakawa, 1976), which is Mercator-type south of 20°N. It has a global horizontal resolution of 1/4°, but is regionally refined between 63°S–10°N and 70°W–70°E via AGRIF two-way nesting (Debreu and Blayo, 2008) to its nominal resolution of 1/20°. This yields an effective resolution of 5.6–27.8

km in the global base and 2.5–5.6 km in the nest, which allows to fully resolve mesoscale processes (c.f., Hallberg, 2013). In the vertical it is composed of 46 z-levels, with grid spacing increasing from 6 m at the surface to maximum 250 m at depth. Bottom cells can be partially filled, thus allowing for an adjustment to a more realistic topography, which has been



obtained from the bathymetry developed by the DRAKKAR community (Barnier et al., 2006) and from interpolating ETOPO1 [https://sos.noaa.gov/datasets/etopo1-topography-and-bathymetry/] for the 1/4° and the 1/20° grid, repectively.

For the employed hindcast (experiment identifier KFS044), the model was initialized with temperature and salinity fields from the Levitus World Ocean Atlas 1998 (Levitus et al., 1998, https://www.esrl.noaa.gov/psd/); spun up from rest for 30 years; and subsequently run with forcing from the interannually varying atmospheric fields of the Coordinated Ocean-Ice Reference Experiments data set version 2 (CORE; Large and Yeager, 2009; Griffies et al., 2009) for the period 1958–2009. Turbulent air-sea fluxes were calculated during the model integration through bulk formula using the prescribed atmospheric state, as well as the simulated sea surface temperatures and relative winds. Lateral boundary conditions were set to free slip in the base model and no slip in the nest. The momentum equations were discretized using the energy and enstrophy conserving (EEN; Arakawa and Hsu, 1990) advection scheme with Hollingsworth correction (Hollingsworth et al., 1983; Bell et al., 2017) and a bi-Laplacian lateral diffusion operator. The evolution of tracers was simulated using the total variance dissipation (TVD; Zalesak, 1979) advection scheme and a Laplacian isoneutral diffusion operator. Viscosity and diffusivity coefficients vary horizontally according to the local grid size and are specified via their maximum values Ahm0 and Aht0, set to $-6 \times 10^9$ ($-1.5 \times 10^{-11}$) $\mathrm{m}^4\mathrm{s}^{-1}$ and 60 (300) $\mathrm{m}^2\mathrm{s}^{-1}$ in the nest (base), respectively. Vertical sub-grid scale physics have been parameterized using a turbulent kinetic energy (TKE; Gaspar et al., 1990; Madec et al., 1998) dependent closure scheme.

In this study we used the simulated 5-day mean velocity and tracer fields from the nested domain, which show a realistic representation of the mean flow pattern and mesoscale eddy activity in the South Atlantic and adjacent Southern Ocean sector (Fig. 1). Most importantly for this study, they adequately represent the major current systems providing the Pacific-Atlantic and Indo-Atlantic inflow of upper limb waters, that are, the ACC and Agulhas Current systems. The mean (2000-2009) transport of the ACC through Drake Passage amounts to $116.2 \pm 7.5$ Sv (1 Sv $:= 10^6$ $\mathrm{m}^3\mathrm{s}^{-1}$, uncertainties are given in terms of one standard deviation). This value is not too far from the ACC transport of $134 \pm 11.2$ Sv reported by Cunningham et al. (2003), given that they estimated the error of this average transport to be between 15 and 27 Sv. Notably, the model features a distinct South Atlantic Current (SAC) north of the ACC and captures the anticyclonic circulation of the Zapiola Gyre (Fig. 1a), while coarser resolution models generally failed to separate the SAC from the ACC and did not resolve the Zapiola Anticyclone (cf. Stramma and England, 1999). In an accompanying study Schwarzkopf et al. (in preparation) show that the total simulated Agulhas Current transport compares reasonably well with available observational estimates from the ACE (at 32°S, Bryden et al., 2005) and ACT (at 34°S, Beal et al., 2015) arrays. Moreover, the simulated standard deviation of sea surface height ($\sigma_{SSH}$), which is a measure of surface geostrophic eddy variability, agrees well with the observed pattern and magnitude of $\sigma_{SSH}$ derived from the Archiving, Validation, and Interpretation of Satellite Oceanographic (AVISO) data product (Fig. 1b-c). In particular, it features the characteristic pattern of high $\sigma_{SSH}$ at the Brazil-Malvinas-Confluence zone, as well as in the extended Agulhas Current system.

Within the simulated 5-day mean velocity fields, also the strength and variability of NBC and AMOC are well represented. Figure 2a shows a meridional velocity section at 6°S as well as the derived timeseries of the simulated NBC transport, obtained from the integration of the northward velocities between 0–1200 m depth and from the coast to 33.5°W (cf. Hummels et al., 2015; Rühs et al., 2015, and references therein for a justification of the adopted NBC definition). One can identify the northward



directed NBC with a subsurface maximum, its southward recirculation to the east, and the southward flowing Deep Western Boundary Current underneath. The mean (2000–2009) NBC transport of $25.5 \pm 5.0$ Sv falls well into the observed range of $26.5 \pm 3.7$ Sv reported by Schott et al. (2005) based on repeated shipboard sections at 6°S (nine section ensemble with individual measurements between 1990 and 2004). The simulated mean (2000–2009) AMOC strength, defined at each latitude as the

maximum of the meridional overturning streamfunction, yields $14.8 \pm 6.3$ Sv at 6°S. Unfortunately, the lack of observational AMOC estimates hamper a model verifications at this latitude. Yet, first results of the SAMBA array (Meinen et al., 2013) allow for some useful quantitative assessments at 34.5°S. Here, the simulated mean (2000–2009) AMOC transport of $13.6 \pm 5.0$ Sv fits well to the recently published observational estimates of $14.7 \pm 8.3$ Sv (Meinen et al., 2018, based on daily data over the 2009–2017 period, with a $\sim 3$ year gap from Dec. 2010 to Sep. 2013). The decadal variability of NBC and AMOC (Fig.

2c) at 6°S features the characteristics already presented in Rühs et al. (2015): the NBC and AMOC transports vary roughly in phase with overall decreasing transports in the 1960s and 1970s, and a recovering in the 1980s and the early 1990s, whereby the decadal NBC variability is more than twice as large in magnitude. This suggests that the conclusions of Rühs et al. (2015) also hold for this model simulation, i.e., that the basin-scale decadal variability of the AMOC is indeed captured in the NBC, but is additionally superimposed (and thus masked) by wind-driven gyre-variability.

## 15   2.2   Offline Lagrangian analysis of AMOC upper limb pathways with ARIANE

We used the ARIANE tool (version 2.2.6, www.univ-brest.fr/lpo/ariane, Blanke and Raynaud, 1997; Blanke et al., 1999) to perform three sets of offline Lagrangian experiments, one reference set (hereafter REF), and two sensitivity sets corresponding to a weak and a strong phase of Agulhas leakage (hereafter lowAL and highAL).

    ARIANE is a freely available FORTRAN software that infers Lagrangian particle trajectories from simulated three-dimensional

volume-conserving velocity fields saved on a C-grid by advecting virtual fluid particles along analytically computed stream-lines. The obtained trajectories thus represent volume transport pathways, which may experience along-track tracer and density changes. For a detailed discussion of this concept please refer to van Sebille et al. (2018).

    For REF we released virtual fluid particles every 5 days for the years 2000–2009 over the full depth of the northward flowing NBC (0–1200 m depth, coast to 33.5°W) at 6°S, yielding a set of 10 one-year release experiments with a total of

O($10^6$) particles. The number of particles seeded at each time step was proportional to the current NBC transport. Following Blanke et al. (1999) each particle was tacked with a partial volume transport (max. 0.01 Sv), so that the cumulative transport of all particles released at each time step reflects the current total NBC transport. Subsequently, the particles were advected backwards in time until the point where they entered the study domain through one of the predefined source sections displayed in Fig. 3, but at maximum for 40 years. During the trajectory integration the potential temperature and salinity fields were

linearly interpolated onto the particle positions.

    For the Lagrangian sensitivity experiments we followed the same procedure as for REF, but used only velocity fields from preselected periods of weak (1960–1969) and strong (2000–2009) Agulhas leakage (based on the Agulhas leakage transport timeseries inferred from a complementary set of offline Lagrangian experiments following Durgadoo et al. (2013), cf. Appendix A). That is, we released particles in the NBC at 6°S for years 1960–1969 (lowAL) and 2000–2009 (highAL), respectively, and





then traced them backwards towards the predefined source sections by looping through the velocity data of each period for at maximum 40 years (instead of making use of the whole simulation period 1958–2009). Even though this looping technique has already been employed by various authors (e.g., Döös et al., 2008; Rühs et al., 2013; Thomas et al., 2015; Berglund et al., 2017; Drake et al., 2018), the obtained results have to be interpreted with caution. Looping may introduce unphysical jumps

in the velocity and tracer fields and, consequently, also in the volume transport pathways and along-track tracer changes. Döös et al. (2008), and Thomas et al. (2015) showed that the errors in the pathways introduced by looping can be negligible if a sufficiently high number of virtual fluid particles is considered and the (model) drift in the velocity fields is not too large. We are accounting for that by continuously seeding particles over 10 years and looping over a period of only 10 years. However, the analysis of along-track thermohaline property modifications makes little sense for timescales exceeding the period of available

data. Therefore, in this study, we employ the looping technique only for estimating sensitivities of the derived volumetric connectivity measures to different idealized states of the South Atlantic circulaton pattern, but not for assessing along-track property modifications.

To quantitatively evaluate the major pathways of the large set of individual trajectories we followed Blanke et al. (1999) and calculated Lagrangian transport streamfunctions (Fig. 3a). These represent time-integrated mean volume transport pathways

derived from all trajectories entering and leaving the domain — particles still in domain were not considered (including them would have violated the constraint of volume conservation). Negative and positive values represent anticyclonic and cyclonic circulation pattern, respectively. A bundling of streamlines highlights most prominent pathways, closed streamlines indicate recirculation pattern.

Note that even though trajectory integrations were performed backwards in time to identify the sources for the upper limb

of the AMOC, in the following the resulting Lagrangian connectivity measures will be described in the more intuitive forward sense, that is, in flow direction.

## 3   Results and discussion

This section starts with a thorough assessment of the Lagrangian trajectory set REF — representing the mean upper limb AMOC connectivity between the Indian/Pacific Oceans and the NBC over the last decades — in terms of a volumetric de-

composition (Sect. 3.1), transit times (Sect. 3.2), volume transport pathways (Sect. 3.3), and along-track thermohaline property modifications (Sect. 3.4). It closes with a short discussion of potential decadal changes in the derived connectivity measures associated with an increase in Agulhas leakage by a joint evaluation of the two trajectory sets lowAL and highAL (Sect. 3.5).

### 3.1   Volumetric decomposition of NBC and AMOC upper limb transport at 6°S

From the Eulerian mean (2000–2009) NBC transport of 25.5 Sv, 2.4 Sv were identified as meanders within the framework

of the Lagrangian analysis. Meanders consists of those particle trajectories that enter and leave the NBC section within one Lagrangian integration time-step of 5 days. Subtracting the meander associated transport yielded a new corrected Lagrangian mean (2000–2009) NBC transport estimate of 23.1 Sv (cf. Fig. 2b), from which 8.7 Sv (38 %) stem from the Equatorial



Atlantic (EQ), 6.3 Sv (27 %) from the Agulhas Current (AC), and 4.7 Sv (20 %) from Drake Passage (DP) (Fig. 3b). Other inflow sections from the Indian Ocean (nIO and eIO) constitute only a minor source of 0.8 Sv. On average 2.6 Sv could not be unequivocally attributed to one of the predefined sources, since the associated particles did not leave the study domain within the 40 years of integration (see discussion below).

If one examines the individual Lagrangian experiments of REF with 10 different release years, one detects a large interannual variability of the total NBC transport as well as in the individual volumetric contributions of the different sources, which necessitates the adopted strategy of multiple Lagrangian experiments with different release periods (Fig. 3c). Most interannual volume transport variability in the NBC is related to variability in the EQ contribution, whereas interannual variability in the AC and DP contributions are less pronounced. Even though the order of the individual sources according to their total volume

transport contribution does remain constant in all experiments and the EQ contribution is always the largest, the relative EQ contribution indeed varies between 31 % and 43 %. Interestingly, the relative AC contribution nearly constantly increases from 25 % in 2000 to 29 % in 2009 (not shown). This increase in the AC contribution could be regarded as an indication that the increase of Agulhas leakage is indeed projected onto the NBC and upper limb of the AMOC in the tropical Atlantic. However, the here discussed Lagrangian analyzes are not sufficient to unequivocally determine a link between Agulhas leakage

strength and the AC contribution to the NBC transport. In principle, a larger Agulhas contribution could stem from a preceded increase in Agulhas leakage and/or from a higher percentage of Agulhas leakage waters reaching the NBC due to changes in the subtropical gyre circulation — even if Agulhas leakage itself stays constant or decreases (cf. Tim et al., in review). In Sect. 3.5, we discuss the possible impact of an ongoing increase in Agulhas leakage on the NBC and upper limb of the AMOC in more detail by making use of lowAL and highAL.

Some uncertainty in our analysis arises from the fact that on average 2.6 Sv of NBC waters could not be sampled at one of the source sections after 40 years of trajectory integration since the corresponding particles remained inside the predefined domain. At the end of the integration period most of these particles could be located within the interior of the tropical and subtropical gyres (not shown), indicating that (multiple) recirculations prevented them from leaving through one of the source sections. To obtain a first estimate of where these waters may stem from, we extended the integration period for the respective

particles to 80 years by cycling through the available velocity data for the period 1958–2009 (similar procedure as for the Lagrangian sensitivity experiments lowAL and highAL described in Sect. 2.2, but by making use of the whole velocity data set instead of restricted time periods). In the additional 40 years of integration, out of the 2.6 Sv, 0.8 Sv could be sampled at the DP, 0.4 Sv each at the AC and EQ, and 0.1 Sv at the IO sections, 0.8 Sv still did not reach any section. This suggests that our analysis of the REF experiments with 40 years integration most likely slightly underestimates the AC, EQ, and in particular

the DP contribution (further support for this assumption follows below in Sect. 3.2).

Of particular interest for this study are those sources contributing to the northward flow of the upper limb of the AMOC, which are the AC, DP, and IO contributions that amount to 11.8 Sv when averaged over all Lagrangian experiments in REF. The EQ contribution is not considered part of the net northward upper limb flow at 6°S, since the southward inflow happens across the same zonal section and in the same depth range as the northward NBC outflow. We attribute this contribution to

the more local tropical circulation, acknowledging that the NBC is not only part of the basin-scale AMOC but also of the



wind-driven horizontal gyre circulation and the shallow overturning of subtropical-tropical cells (Rühs et al., 2015). Adding the 1.3 Sv that according to the extended experiments with 80 years of integration most likely also stem from AC, DP, and IO, yields an estimate for the Lagrangian mean upper limb transport at 6°S of 13.1 Sv. This number compares well with the simulated mean (2000–2009) Eulerian AMOC strength at 6°S of $14.8 \pm 6.3$ Sv, which gives us confidence in the validity of our

Lagrangian decomposition of the AMOC's upper limb with the NBC at 6°S as a reference point. The remaining discrepancy between the Lagrangian and Eulerian estimate of the AMOC strength can be attributed to (i) particles that did not cross any sampling section but remained in the domain even after 80 years integration, in total amounting to 0.8 Sv, (ii) a minor upper limb contribution flowing northward in the interior tropical Atlantic (additional Lagrangian experiments backwards in time from the EQ instead of the NBC section yield quantitative estimates for that contribution of 1.0 and 1.2 Sv after 40 and 80

years of integration, respectively), and (iii) differences in Lagrangian and Eulerian averaging.

Our results — with and without the second integration cycle — seem to generally agree with the prevailing conception that the warm water route is the major source for the return flow. The respective AC and minor IO contribution to the AMOC's upper limb transport at 6°S amounts to 60 % (7.1 from 11.8 Sv) in REF, and to 58 % (7.6 from 13.1 Sv) in the extended experiment. Nonetheless, we also obtained a DP contribution of 40 % (4.7 from 11.8 Sv) and 42 % (5.5 from 13.1 Sv), respectively,

which, to our knowledge, is substantially higher than those inferred from all previous Lagrangian model studies, but now better matches those derived from observations. For instance, with regard to Lagrangian model studies, Speich et al. (2001), Donners and Drijfhout (2004), and Speich et al. (2007) only estimated a direct cold water route contribution of 13 % (2.3 from 17.8 Sv at 20°N), 6 % (1.0 from 16.2 Sv at 0°N), and < 6 % (< 1.0 from 17.4 Sv at 44°N), respectively. A possible reason for our remarkably different rating of the relative importance of the direct DP contribution could be the increased horizontal (and

temporal) resolution of the underlying model data of this study compared to that of the previous ones. On the one hand, an increased resolution leads to a way more realistic representation of the current structure in the southern South Atlantic. Despite generally enhanced eddy-driven cross-frontal transports it allows for the separation of the SAC from the ACC as described in Sect. 2.1, and a more detailed representation of the intricate flow pattern in the Brazil-Malvinas-Confluence zone. All these features potentially influence the existence and strength of the direct DP contribution (cf. Sect. 3.3). On the other hand, coarser

resolution models are likely to overestimate the AC contribution due to unrealistically high Agulhas leakage (cf. Sect. 1), thereby reducing the relative importance of the DP contribution. It must be underlined however, that Speich et al. (2001) and Speich et al. (2007) used an OGCM in a robust diagnostic mode. Therefore, the analyzed fields have to be interpreted rather as a dynamical interpolation of observations than as a prognostic model output. Hence, more dedicated studies are needed to test the sensitivity of the relative DP and AC contributions to the model resolution, as well as to the potential impact of other details

in the model configuration and forcing. In terms of observations, estimates of corresponding mean absolute transports were not available until Rodrigues et al. (2010) addressed this issue. They used the hydrographic and sub-surface float data from the World Ocean Circulation Eperiment (WOCE) to estimate total transports for key regions in the South Atlantic, yielding a cold-water contribution of 36 % (4.7 from 13.2 Sv at 32°S). Hence, these observation-based estimates support the findings of our new model-based analysis.



Due to the reasoning outlined in Sect. 2.2, for the rest of this study results are based on the REF experiments without cycling of the velocity fields (except for Sect. 3.5, where lowAL and highAL are analyzed). Since the majority of DP and AC particles do reach the NBC within the REF integration period of 40 years (cf. Sect. 3.2), we are convinced that the general scientific interpretation will not change with increasing integration time, even though absolute numbers may do (as demonstrated above).

## 3.2 Transit times towards NBC at 6°S

The transit times of waters from the different sources to reach the NBC are crucial for understanding how these source waters and potential changes therein are transported downstream and thus impact the mean characteristics and variability of NBC waters and the upper limb of the AMOC.

Figure 4 shows the transport-weighted distributions of the advective transit times through the South Atlantic from the EQ, AC, and DP sections, respectively: From the EQ section the NBC is reached by the majority of particles (> 50 %) in less than 3 years, with 1 year being the most frequent transit time (modal value of the transit time distribution), which results from short pathways on which changes can be relatively fast and directly transmitted. This is one reason why interannual variability in the NBC mostly corresponds to variability of the EQ contribution, which is most probably a response to local variability in the wind forcing (cf. Rühs et al., 2015). From the AC and particularly DP the majority of particles (> 50 %) reach the NBC in 9 years and 18 years, and the most frequent transit times are 7 years and 12 years, respectively, hence they need considerably more time. Additionally to the shift to longer time scales the transit time distributions are broadening, in particular for waters with DP origin, representing longer and more diverse connecting pathways. The interannual variability of the volumetric AC and DP contributions thus constitute the accumulation of past changes at the source and potential additional circulation changes along the way. This may impede a direct imprint of variability at the DP and AC sources on the NBC.

The derived transit time distributions for the AC contribution closely match those derived by Rühs et al. (2013) based on output from the ocean model configuration INALT01, the precursor of the here employed INALT20. As already stated in Rühs et al. (2013), the transit times appear slightly longer than those estimated by van Sebille et al. (2011), even though a detailed comparison is inhibited by differences in the applied methodologies and emphases of the studies.

Our derived transit times from the DP into the equatorial Atlantic can be compared to the transit times from DP towards 20°N estimated by Speich et al. (2001). Most notably, Speich et al. (2001) arrived at a multi-modal distribution and interpreted peaks at 19 years and 29 years as the representative time periods needed for DP waters to reach the North Atlantic on a relatively direct path and with one recirculation in the South Atlantic, respectively. The 19 years for the direct path are comparable to our 12 years, if following Rühs et al. (2013) and assuming the transit from 6°S to 20°N to account for another ∼ 6 years. However, our analysis does not show a second peak. This may be due to differences in the horizontal and temporal resolution of the employed ocean models. Our analysis is based on a high-resolution fully eddying configuration, whereas Speich et al. (2001) used a coarse, non-eddying, one. Moreover, we used 5-day mean velocity fields of a hindcast experiment forced with interannually varying atmospheric fields, whereas Speich et al. (2001) used monthly means from a climatological experiment. The increase in resolution and allowance for interannual variability most likely lead to more diverse recirculation pathways and associated transit times, thereby disintegrating the second peak in the transit time distribution.





Further note that the shape of the transit time distribution for the DP contribution shows a less distinct modal value and a broader tail than those for the AC and EQ distributions. This supports our assumption (cf. Sect. 3.1) that a large part of the particles still in the domain after 40 years of integration probably could be attributed to the DP source and consequently reinforces the importance of the DP as a contributor to the upper limb of the AMOC.

## 3.3 AMOC upper limb pathways in the South Atlantic

In this section we have a closer look at the mean AMOC upper limb pathways in the South Atlantic by investigating the connection from the two major individual sources, that are the AC and DP regions, towards the NBC. This is done by the use of conditional Lagrangian streamfunctions, which were calculated considering only the respective subsets of trajectories and represent the associated net advective volume transport pathways (Fig. 5).

From the Indian Ocean the dominant volume transport pathways towards the NBC are via the narrow Agulhas Current, Agulhas leakage — in form of Agulhas rings or within the Benguela Current — and the broad South Equatorial Current (Fig. 5a), as already described in previous studies (e.g., van Sebille et al., 2011; Rühs et al., 2013). Some recirculation may occur in the subtropical gyre ($\sim 1$ Sv) and in the Agulhas basin itself ($\sim 3$ Sv). AC waters contributing to the NBC enter as surface and intermediate waters over the whole longitudinal and depth range of the Agulhas Current and later occupy the whole longitudinal and depth range of the NBC (Fig. 6a,c).

From the Pacific, the majority of fluid particles later reaching the NBC through a relatively direct path (without a detour through the Indian and eventually also Pacific Ocean within the Southern Hemispere supergyre) is entering the Atlantic through the northern part of Drake Passage ($\sim 3$ Sv), then follows a narrow path through the Malvinas Current and the South Atlantic Current, before entering the South Equatorial Current over a broad longitudinal range in the eastern part of the basin (Fig. 5b) as roughly described in Speich et al. (2001). However, as already indicated by the relatively broad transit time distribution, there are many different pathways for DP waters towards the NBC with different side-tracks: Some particles amounting to $\sim 1$ Sv follow the ACC into the eastern part of the basin, eventually make a little detour into the Agulhas basin and finally enter the South Equatorial Current via the Benguela Current System. $\sim 1$ Sv of DP waters is recirculated in the South Atlantic subtropical gyre at least once before finally entering the NBC.

A comparison of the DP particle distributions at the NBC section with the local Eulerian mean cross-section velocities shows that DP particles can be found over the whole depth range and zonal extent of the NBC roughly proportional to the local current strength (Fig. 6d). The DP particle distribution at the source section, however, does not directly reflect the Eulerian cross-section flow through Drake Passage (Fig. 6b). The eastward ACC transport through Drake Passage is known to occur mainly within two jets corresponding to the Subantarctic Front and the Polar Front (Cunningham et al., 2003; Firing et al., 2011). These two maxima are captured in the model simulation, but only from the northern one virtual fluid particles may directly enter the subtropical gyre and contribute to the South Atlantic branch of AMOC's upper limb. Particles that follow the ACC south of 53°S eastward beyond 35°W have zero chance to enter the subtropical gyre before leaving the Atlantic sector of the Southern Ocean south of Africa towards the Indian Ocean sector (at least in this model study). These imaginary borders also define the separation between the direct and indirect cold water route, which in our study is located more eastern than



reported by Speich et al. (2001) who identified it at about 50°S and 50°W. This difference is most probably again related to more diverse pathways in our higher resolution model simulation.

When comparing the Lagrangian streamfunctions and depth distributions from the AC and DP contributions, it becomes apparent that the dominant spreading pathways of both contributions coincide in the horizontal as well as in the vertical plane

within the subtropical gyre. However, the spreading of AC waters is concentrated in slightly more northern and shallower branches of the SEC than the spreading of DP waters. Consequently, within the NBC, AC waters dominate in the upper 400 m and DP waters below. The coinciding pathways alreday indicate that both contributions potentially mix along their transit through the South Atlantic and may experience thermohaline property changes.

### 3.4   AMOC upper limb water properties modification within the South Atlantic

Figure 7a-d visualize the mean potential temperature ($\theta$) and salinity (S) characteristics of DP and AC waters at their respective source and within the NBC by means of histograms of Lagrangian particle frequency in $\theta$-S space (0.5 °C × 0.05 bins). In their source regions, the DP contribution is relatively fresh (32.25 to 34.75) and cold ($-2$ to 11.0 °C) with potential density anomalies ($\sigma_\theta = \rho_\theta - 1000$ kgm$^{-3}$, in the following units are dropped for better readability) between 25.9 and 27.8, whereas the AC contribution is more salty (34.30 to 35.75) and spans a broader temperature range (2.5 to 29.0 °C) with $\sigma_\theta$ between 21.7

and 27.7. Note that there is no clear separation of the two sources in temperature as the terminology *cold* and *warm* water routes may imply, since the AC contribution does not only consist of warm surface waters, but also of colder central and intermediate waters (Beal et al., 2006; Speich et al., 2007; Biastoch and Böning, 2013). In our Lagrangian model analysis 99 % of the DP (AC) contribution originally has temperatures colder (warmer) than 8.5 °C (4.0 °C), and 75 % (11 %) of DP (AC) waters can be found in the respective shared temperature range from 4.0 °C to 8.5 °C. However, the AC and DP water contributions can

be well distinguished by the salinity in their source regions: 99 % of the DP (AC) contribution originally has salinities lower (greater) than 34.45. Thus we may consider *fresh* and *salty* routes as an alternative more precise terminology, even though we would still recommend referring directly to the geographic origin to avoid ambiguities. Upon arrival in the NBC, fluid particles with DP origin cover nearly the same $\theta$-S spectrum as those particles with AC origin, that are, temperatures from 3.5 °C to 30.0 °C and salinities from 34.25 to 37.60 (34.25 to 37.65 for AC waters), with corresponding $\sigma_\theta$ from 22.7 to 27.6 (23.0 to 27.6 for

AC waters). The comparison of the initial and final $\theta$-S spectra shows that substantial thermohaline property modification does occur on the transit through the South Atlantic. The bulk of waters entering the South Atlantic through AC and DP that later reach the NBC becomes more salty during the transit. Waters of DP origin additionally experience a substantial broadening of their temperature spectrum associated with a general warming.

These thermohaline property modifications are also eventually associated with transformations in density space. To quantify

those, we binned the partial transports of all particles according to their potential density anomaly at their origin (Fig. 7e) and upon arrival in the NBC (Fig. 7g), for each contribution separately. Then we compared the transport-weighted density distributions at the entry sections and the respective distributions in the NBC. The difference between the initial and final distributions quantifies the net transformation of the bulk of water (Fig. 7f), which may be a result of multiple modification processes. We additionally repeated this analysis for respective subsets of surface ($\sigma_\theta < 26.0$), central ($26.0 < \sigma_\theta < 27.0$),





and intermediate ($27.0 < \sigma_\theta < 27.5$) waters. The exact numbers for the transformation analysis of the different classes depend on the applied density criteria. Here we chose to follow Macdonald (1993) and Holfort and Siedler (2001) for the definition of the separation of central and intermediate waters, since it fits to the simulated mean (2000-2009) thermohaline structure of the water column in the South Atlantic (Fig. 8). Other authors following Roemmich (1983) chose $\sigma_\theta =26.8$, but in our

simulation waters with $26.8 < \sigma_\theta < 27.0$ seem to fall in the central water range with a still relatively large vertical gradient in temperature and salinity. The general tendencies of our transformation analysis stay robust with respect to small changes in these criteria (not shown), so that the results are worth a discussion. The AC contribution to the upper limb of the AMOC originally consisted of 2.0 Sv surface, 3.7 Sv central, and 0.6 Sv intermediate waters. These waters showed a net transformation to 1.2 Sv surface, 4.4 Sv central, and 0.7 Sv intermediate waters upon arrival in the NBC. More specifically, 40 % (0.8 Sv)

of the surface waters became central waters, 8 % (0.3 Sv) of the central waters became intermediate waters, and and a third (0.2 Sv) of the intermediate waters became central waters (Table 1). Hence, the bulk of AC waters experiences a slight net density gain during the transit towards the NBC. The DP contribution to the upper limb of the AMOC originally consisted of 1.6 Sv central, and 3.1 Sv intermediate waters. These showed a net transformation to 0.9 Sv surface, 2.3 Sv central, and 1.5 Sv intermediate water upon arrival in the NBC. More specifically, 31 % (0.5 Sv) of central waters became surface waters (6 %

became intermediate waters), and 55 % (1.7 Sv) of the intermediate waters became central or surface waters (Table 1). To sum it up, in contrast to AC waters, the bulk of DP waters experiences a net density loss during the transit towards the NBC.

Changes of temperature and salinity and thus density along volume transport pathways generally correspond to the mean effect of surface fluxes, such as direct warming by solar heat flux or precipitation and evaporation processes, and mixing with ambient waters due to parameterized and spurious tracer diffusion of the OGCM. The fact that both DP and AC waters show a

salinification during their transit implies that they do not only mix with each other, but that mixing with other ambient waters and/or surface fluxes may play an important role, too.

To further investigate the transformation in the bulk AC and DP water volumes occurring during the transit and to determine their origin, we assessed which particles reached the mixed layer on their transit between the DP or AC and the NBC, and then investigated the net transformation of waters with and without mixed layer contact separately. Following the criterion

used during the OGCM integration and also adopted in comparable Lagrangian studies (cf. Blanke et al., 2002; Tim et al., in review), we assume a particle having reached the mixed layer if its density (that equals the ambient density) differs by less than $0.01 \ \mathrm{kgm^{-3}}$ from the density at 10 m depth. During the transit through the South Atlantic the surface mixed layer is reached at least once by 66 % (3.1 from 4.7 Sv) and 49 % (3.1 from 6.3 Sv) of DP and AC waters, respectively. That implies that these waters most likely do gain their specific characteristics within the South Atlantic and — if adopting the common definition of

a water mass as a body of water with common formation history — are strictly speaking not water masses with Indian Ocean or Pacific origin as may be suggested by their prior classification as AC or DP waters.

Fig. 9 maps the frequency distribution of the position of the last mixed layer contact for particles on the transit between AC (left) or DP (right) and the NBC. Regions of high frequency can be interpreted as the most probable formation regions for water masses of the NBC within the South Atlantic. Those can be found in the vicinity of different mode water formation regions as

depicted in Hanawa and Talley (2001, hereafter H01) and Sato and Polito (2014, hereafter S14): in the eastern and southern





subtropical gyre where eastern subtropical mode water (ESTMW or SASTMW2 in H01 and S14, respectively) and southern subtropical mode water (SASTMW3 in S14) are formed, respectively, as well as east of Drake Passage where subantarctic mode water (SAMW in H01) originates. Notably, the formation region of classical western subtropical mode water (STMW or SASTMW1 in H01 and S14, respectively) located east of the western boundary current, that is, the Brazil Current, does

not stand out in the frequency distribution. That suggests that western subtropical mode water is no major contributor to NBC waters in the analyzed model simulation. A possible explanation for this could be that most western subtropical mode water gets re-entrained into the mixed layer in the eastern subtropical gyre and thus becomes itself part of the eastern subtropical mode water. In any case, our findings match those of Tim et al. (in review), who analyzed the same model simulation and found an only minor direct South Atlantic Central Water but strong eastern South Atlantic Central Water contribution to the upwelling

water masses of Benguela. Relative high frequencies of last mixed layer contacts are further located at the subduction zones along the South Equatorial Current and in particular around 15°S, which have been shown to be important source regions for the equatorward subsurface flow related to the shallow overturning of the subtropical cell in the South Atlantic (Zhang, 2003; Schott et al., 2004; Hazeleger and Drijfhout, 2006).

The mixed layer contact of virtual fluid particles on their transit between the AC or DP and the NBC greatly impacts

their characteristic properties. Most notably, the net transformation in density space of DP and AC waters (Fig. 7f) is almost completely captured by those particles with mixed layer contact (Fig. 10d), whereas water property modifications associated with particles without mixed layer contact are characterized by only minor changes in density space (Fig. 11d).

Nearly all DP particles entering the South Atlantic in the central water range and more than half (1.6 from 3.1 Sv) of those entering in the intermediate water range reach the mixed layer at least once during their transit. The bulk of these DP waters

with mixed layer contact experience a substantial salinification (when entering the Atlantic 95 % of the waters have salinities < 34.45, whereas upon arrival in the NBC 95 % of the waters have salinities > 34.45) but even stronger warming during the transit. This results in a net decrease in density and in a corresponding shift towards central and surface waters. From the AC contribution, the particles entering the South Atlantic as surface waters, as well as 30 % (1.1 from 3.7 Sv) of those entering as central waters reach the mixed layer during their transit. They enter the South Atlantic at a large range of relatively high

temperatures which hardly changes during the transit. However, as DP waters, AC waters with mixed layer contact experience a net salinification, resulting in a net increase in density. Since the coherent increase in salinity of AC and DP waters can not be found for waters without mixed layer contact, we attribute the salinification to the impact of on average net evaporative surface fluxes over the subduction zones along the South Equatorial Current, as well as to mixing with ambient subtropical upper layer waters with high salinities.

Nearly half (1.5 from 3.1 Sv) of the DP waters that are entering the South Atlantic as intermediate waters, as well as nearly all intermediate and 70 % (2.6 from 3.7 Sv) of the central AC waters do not experience any mixed layer contact during their transit towards the NBC. Interestingly, AC and DP waters without mixed layer contact show opposite transformations in temperature and salinity: DP waters are warming and becoming more saline, whereas AC waters are cooling and freshening. As a result the salinity and temperature distributions of both components show new common peaks in the temperature and salinity

distributions, at 3.5–4.0 °C and 34.35–34.40, respectively. This result fits to the idea that intermediate and central waters of



western and eastern Atlantic origin mix in the eastern South Atlantic and form new varieties of central and intermediate water
(water masses in the same density range but with different temperature and salinity characteristics), as specifically shown by
Rusciano et al. (2012) from observations, and Rimaud et al. (2012) from a regional modelling experiment for different varieties
of Antarctic Intermediate Water.

## 3.5 Potential decadal changes in AMOC's upper limb connectivity measures

Figure 12 gives a first impression on the possible sensitivity of the volumetric NBC decomposition to the strength of Agulhas
leakage. The underlying data stem from the two additional sets of Lagrangian experiments for which particles were released in
the NBC at 6°S for years 1960 to 1969 (lowAL) and 2000 to 2009 (highAL), respectively, and then traced backwards towards
the source sections (cf. Fig. 3) by cycling through the velocity data of each period for at maximum 40 years (more details on
the experiments can be found in Sect. 2.2).

We chose the 1960s and 2000s for our comparison, since both periods feature the same simulated Lagrangian mean NBC
transport (23.1 Sv) and comparable upper limb transport (11.0 Sv in the 1960s and 12.0 Sv in the 2000s) estimates, but are
associated with notably different estimated mean values of Agulhas leakage that show an increase from 8.4 Sv in the 1960s to
around 14.6 Sv in the 2000s (note that the mean Agulhas leakage transport for the 2000s has been calculated based only on
the annual mean values from 2000 to 2005, since the applied methodology of Agulhas leakage estimation required a potential
particle tracking towards the Good Hope section for at least 4 years and the employed model simulation only provided velocity
output until 2009, cf. Appendix A). The fundamental question is whether this increase in Agulhas leakage also evoked a
corresponding increase in the AC contribution to the upper limb of the AMOC.

In highAL the AC contribution amounts to 7.0 Sv which constitutes 30 % of the total NBC transport and 58 % of the AMOC
upper limb transport at 6°S; whereas the DP only contributes 4.1 SV, that is, 18 % and 34 % to the NBC and upper limb
transports, respectively. In contrast, in lowAL the DP contribution slightly exceeds the AC contribution. The AC contribution
only amounts to 5.1 Sv, that is, 22 % of the total NBC transport and 46 % of the AMOC upper limb transport at 6°S; whereas
the DP contribution provides 5.3 SV, that is, 23 % and 48 % to the NBC and upper limb transports, respectively. In highAL
and lowAL on average 2.1 and 2.8 Sv remained in the domain after 40 years of integration, respectively, from which a large
part probably would additionally add to the DP contribution under longer integration times (cf. Sect. 3.1). The higher amount
of unsampled particles in lowAL can be related to the generally longer transit time from DP to NBC than from AC to NBC: A
circulation pattern with more pronounced DP contribution is associated with a shift of the total transit time distribution, that is,
a transit time distribution inferred of particles from all sources, towards longer time scales (not shown); it consequently yields
more particles still in the domain after 40 years of integration. These estimates suggest that the increase in Agulhas leakage
between the 1960s and 2000s is indeed reflected in an increase of the AC contribution to the AMOC's upper limb transport and
is further accompanied by a decrease in the DP contribution.

The increase in the AC contribution to the AMOC's upper limb is, however, not directly proportional to the increase in
Agulhas Leakage, but weaker (1.9 Sv compared to 6.2 Sv, respectively). This fits to the findings of Durgadoo (2013), who
assessed the impact of Southern Hemisphere wind changes on the strength and fate of Agulhas leakage. He showed that the



simulated wind-driven increase in Agulhas leakage goes along with a strengthening of the South Atlantic subtropical gyre. This strengthening leads to a favored re-circulatory route at the bifurcation point of the South Equatorial Current at the coast off Brazil. Consequently, less Agulhas leakage waters feed into the North Brazil Current. The strengthened re-circulatory route could also be regarded as one potential reason for the simulated decrease in the DP contribution. However, due to the

conceptual limitations of the here presented idealized analysis, the results are not yet conclusive and should be rather regarded as a motivation for future studies.

## 4  Summary and Conclusions

In this study we revisited the relative importance of (i) the relatively warm and salty waters entering the South Atlantic from the Indian Ocean mainly via the Agulhas Current (AC) system, commonly referred to as the warm water route, and (ii) the colder

and fresher waters entering directly from the Pacific through Drake Passage (DP), termed the (direct) cold water route, for the northward volume transport and thermohaline properties of the upper limb of the Atlantic Meridional Overturning Circulation (AMOC).

To do so we used the simulated three-dimensional 5-day mean velocity and tracer fields from a hindcast experiment (1958–2009) with the high-resolution ($1/20°$) ocean general circulation model INALT20; and employed the Lagrangian tool

ARIANE to calculate $O(10^6)$ advective volume transport trajectories as well as along-track thermohaline property changes between the two source regions and the North Brazil Current (NBC), which channels the upper limb flow in the tropics. The main results in terms of connecting pathways, associated volume transports, and major areas of thermohaline property modifications through mixed layer contact are summarized in Fig. 13.

Even though our results generally agree with the prevailing conception that the AC contribution is with around 50 % the

major source for the *upper limb transport* of the AMOC (and yields together with other minor Indian Ocean contributions a total warm water route contribution of 60 %), they also suggest a non-negligible DP or cold water route contribution of at least 40 %. That is a substantially higher value for the DP contribution than previously inferred by Lagrangian studies with coarser resolution models (6-13 %, Speich et al., 2001; Donners and Drijfhout, 2004; Speich et al., 2007), but now better matches estimates from observations (36 %, Rodrigues et al., 2010). Yet, our first idealized analysis of potential decadal changes in the

DP and AC contributions indicates that the ratio of the two sources is subject to temporal variability. On the one hand, there may have existed phases in which DP and AC yielded comparable volumetric contributions to the upper limb of the AMOC; on the other hand, the ongoing increase in Agulhas leakage indeed could have evoked an increase in the AC contribution — though this increase is weaker than the increase in leakage itself and goes along with a decrease in the DP contribution.

In terms of *thermohaline properties*, our study highlights that waters with DP and AC origin cannot be clearly distinguished

by the temperature at their source as the commonly adopted terminology may imply, but instead by their salinity. It further reveals substantial thermohaline property modifications of AC and DP waters during their transit through the South Atlantic, upon arrival in the NBC waters of both origins can be found in the same temperature and salinity range. The bulk of DP waters experiences a net density loss through a net warming, the bulk of AC waters a slight net density gain through a net increase



in salinity. These modifications in density space are nearly completely captured by those Lagrangian particle trajectories that are reaching the surface mixed layer at least once during their transit, which amount to 66 % and 49 % from the DP and AC waters, respectively. This implies that 53 % of the *water masses* supplying the upper limb of the AMOC are formed within the South Atlantic, and only 14 % and at maximum 33 % get their characteristic properties in the Pacific and Indian Oceans,

respectively. This stresses the point, that the South Atlantic is not only a passive conduit for remotely formed water masses, but instead represents an active source and modifier of water masses contributing to the upper limb of the AMOC in the tropics (cf. Garzoli and Matano, 2011).

The substantial along-track property modifications challenge the use of the inferred advective volume transport pathways and timescales for assessing the pathways and timescales with that upper ocean temperature or salinity anomalies are transmitted

through the South Atlantic. They imply that, dependent on the amplitude and duration of the original anomaly, diffusive mixing with ambient waters and surface fluxes will strongly dampen the anomaly along the advective pathways. Overall, this is in line with the argumentation of Lozier (2010) who stated that the view of the global meridional overturning circulation as a coherent conveyer belt transporting heat and dissolved substances through the world ocean is on its way to deconstruction. It is noteworthy though that the deeper parts of the upper limb, such as the intermediate waters transiting the South Atlantic

without mixed layer contact, largely keep their characteristic properties along their transit. Hence, at least for these water types, simulated advective pathways and timescales remain relevant for the propagation of potential temperature and salinity anomalies. However, more work is needed to better understand, how potential temporal changes in the Pacific-Atlantic and Indo-Atlantic exchange are related and eventually projected onto the return flow characteristics, and, in particular, which impact water mass formation and horizontal circulation pattern within the South Atlantic have on these processes. This knowledge

is crucial for a better assessment of the potential impact of the observed warming in the extended Agulhas region and the estimated increase in Agulhas leakage over the last decades.

Finally, even though in this study we focused on the importance of the ratio of the DP and AC contributions for the upper limb of the AMOC, the ratio of the two inflows generally impacts the thermohaline structure of the whole South Atlantic, with consequences also on a more regional scale, for instance, for the Benguela upwelling region (Tim et al., in review) or the

tropical Atlantic (Lübbecke et al., 2015; Castellanos et al., 2017).

## Appendix A: Lagrangian estimation of Agulhas leakage transport timeseries

The timeseries of Agulhas leakage has been inferred by Schwarzkopf et al. (in preparation) from a complementary set of offline Lagrangian experiments with ARIANE in forward mode, following Durgadoo et al. (2013): Virtual fluid particles were released in the Agulhas Current at 32°S every 5 days for the years 1958–2005 proportional to the current volume transport, each

particle associated with a fraction of this transport. Subsequently, the particles were traced for at maximum 5 years towards predefined control sections around the extended Agulhas region. All particles leaving the domain through the so-called Good Hope section (Ansorge et al., 2005) towards the South Atlantic were marked as Agulhas leakage and their individual transports were subsequently combined to estimate a timeseries of Agulhas leakage referenced to the release year of the particles.



*Author contributions.* SR and AB defined the overall research problem and methodology; SS helped to refine research questions and methodology; FUS developed, ran, and validated the OGCM; SR performed and analyzed the main Lagrangian simulations; FUS performed and analyzed the additional Lagrangian experiment to determine Agulhas leakage; SR produced all figures; SR prepared the manuscript with contributions from all co-authors

5  *Competing interests.* No potential conflict of interest was reported by the authors.

*Acknowledgements.* The OGCM and trajectory simulations were performed at the High Performance Computing Center in Hannover (HLRN). The project received funding by the Cluster of Excellence 80 'The Future Ocean' within the framework of the Excellence Initiative by the Deutsche Forschungsgemeinschaft (DFG) on behalf of the German federal and state governments (grant CP1412, SR); by the German Federal Ministry of Education and Research (BMBF) for the SPACES-AGULHAS project (grant 03F0750A, AB and FUS); by the

10  European Union's Horizon 2020 research and innovation programme as part of the AtlantOS project (grant 633211, SS); and by the SAMOC project (grant 11-ANR-56-004, SS). The authors further wish to thank Susan Lozier and Erik van Sebille for inspiring discussions that added value to this manuscript, Bruno Blanke and Nicolas Grima for realising and helping to tackle the Lagrangian ARIANE software, and Willi Rath for technical support.



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

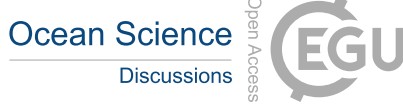

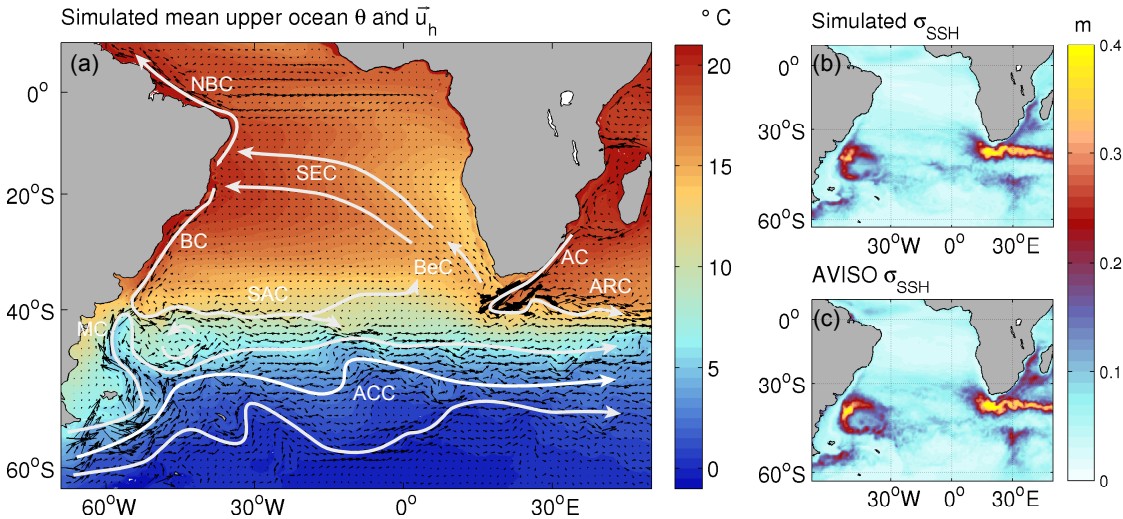

**Figure 1.** South Atlantic circulation pattern. (a) Simulated mean (2000–2009) upper ocean (averaged over upper 750 m depth) potential temperature $\theta$ (color shading) and horizontal velocity $\boldsymbol{u_h}$ (vectors); velocity components have been averaged onto a $1.5° \times 1.5°$ grid for plotting purposes; white arrows highlight major currents of interest for this study: Agulhas Current (AC), Agulhas Return Current (ARC), Benguela Current (BeC), Antarctic Circumpolar Current (ACC), Malvinas Current (MC), South Atlantic Current (SAC), South Equatorial Current (SEC), North Brazil Current (NBC) and Brazil Current (BC). (b-c) Mean standard deviation of sea surface height $\sigma_{SSH}$ from simulations and AVISO, based on 5-day means in 2000–2009.



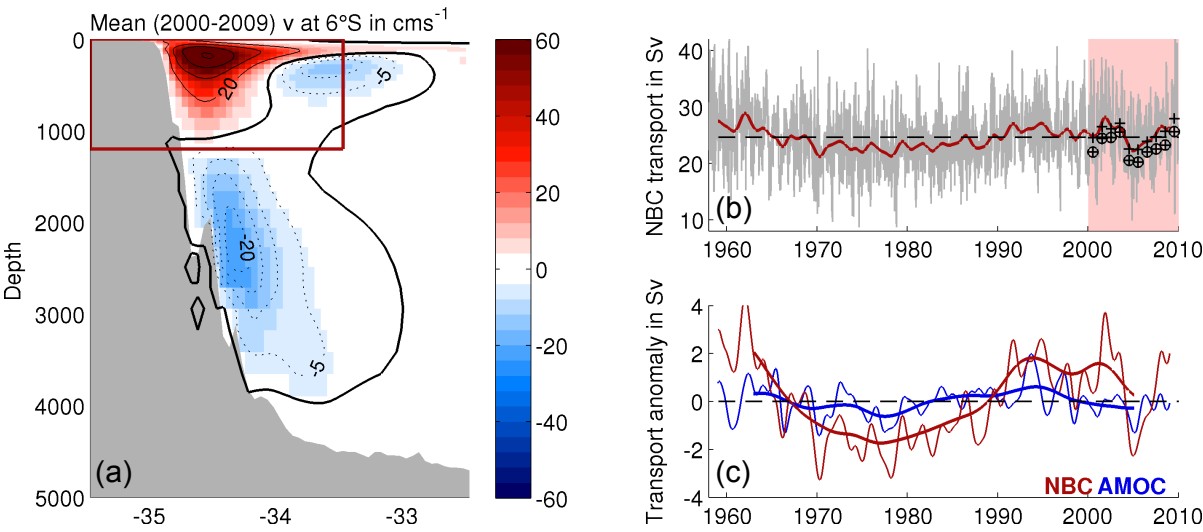

**Figure 2.** Simulated NBC and AMOC at 6°S. (a) Mean (2000–2009) meridional velocity v in the western tropical Atlantic; the NBC transport is defined by the integrated positive meridional (= northward) velocities between the coast and 33.5°W, and 0 –1200 m depth (red box). (b) 5-day mean (grey) and interannually lowpass-filtered (red) NBC transport time series; the release period for the Lagrangian REF experiments is highlighted in light red, annual mean Lagrangian transport estimates without (crosses) and with (crosses in circles) correction for meanders are marked. (c) Interannual (thin) and decadal (thick) NBC (red) and AMOC (blue) transport anomalies (long-term mean subtracted).



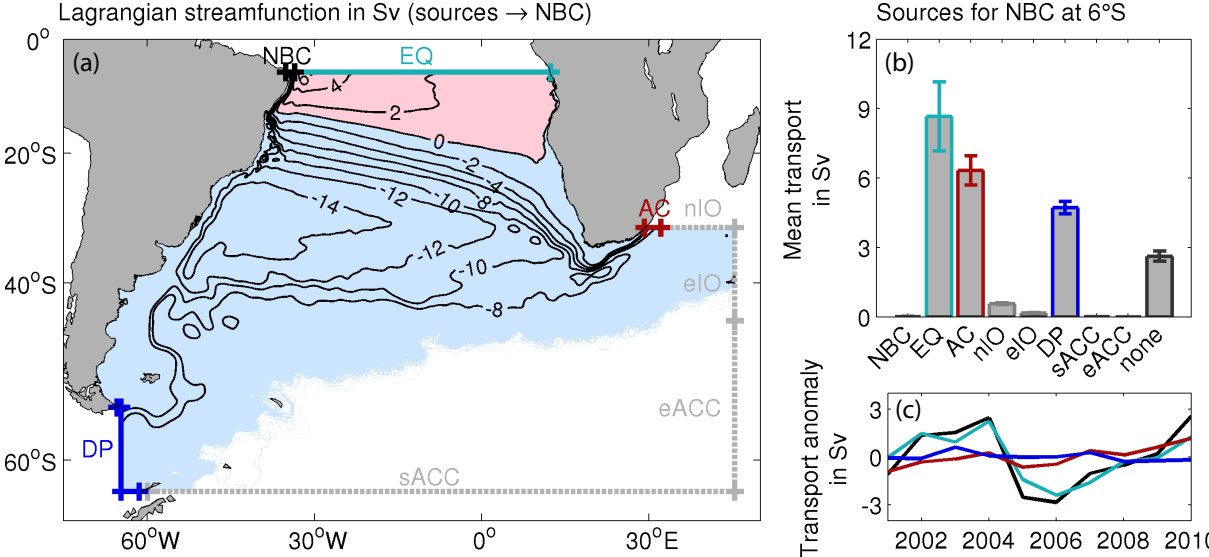

**Figure 3.** Sources for NBC transport inferred from the REF set of 10 Lagrangian experiments for which particles were released in the NBC at 6 °S every 5 days for years 2000 to 2009 and then traced backwards in time towards the indicated source sections for maximum 40 years. (a) Mean Lagrangian streamfunction representing volume transport pathways from all source sections towards the NBC. (b) Mean volumetric contributions of the individual sources to the NBC; whiskers indicate the range of transport estimates. (c) Timeseries of interannual variability of the total NBC transport (black line) and its individual contributions, that are, volumetric contribution of each Lagrangian experiment plotted against the respective release year (colored lines).





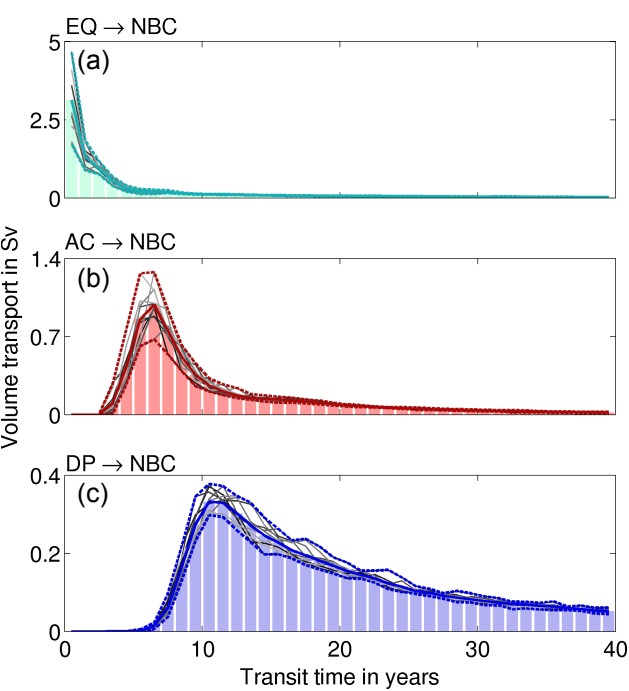

**Figure 4.** Transit times from the (a) EQ, (b) AC, and (c) DP source region towards the NBC at 6°S; bars and solid colored lines represent mean transport-weighted transit time distributions, dashed colored lines indicate the range of the 10 REF experiments (displayed in thin grey solid lines).



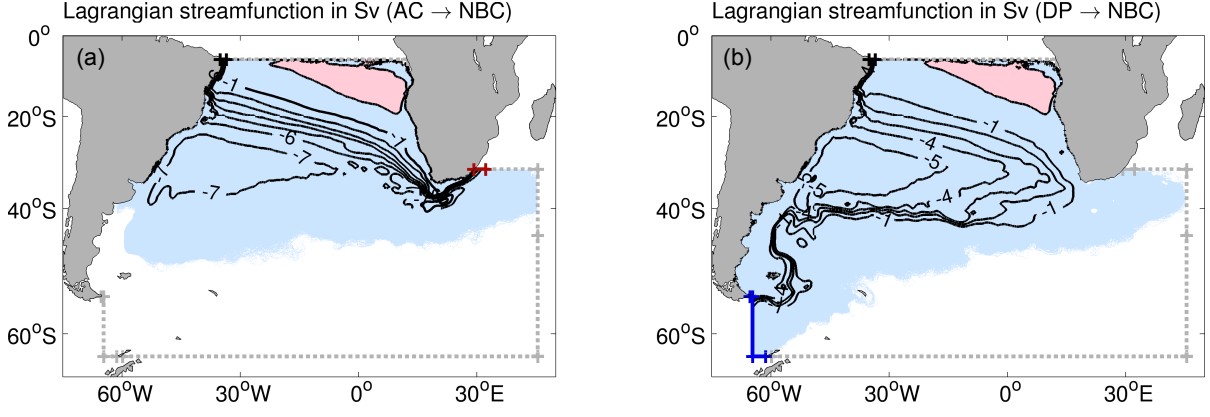

**Figure 5.** Mean Lagrangian streamfunction from all 10 REF experiments representing net volume transport pathways between (a) AC, or (b) DP and the NBC; contour intervals of 1 Sv.





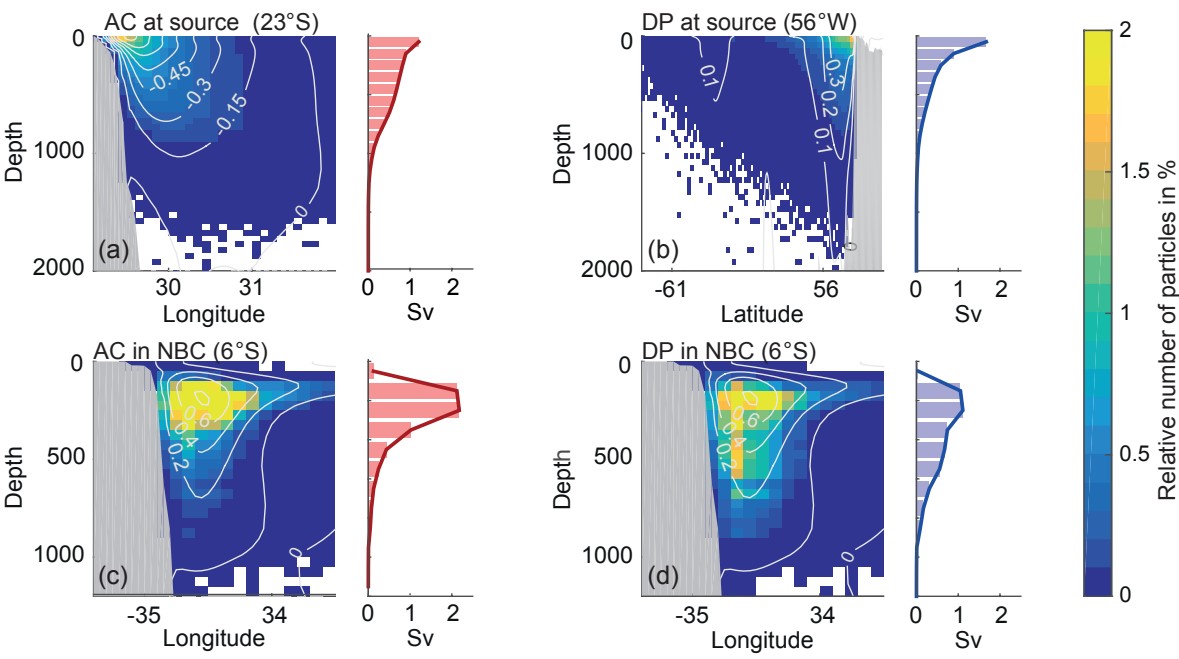

**Figure 6.** Depth distribution for (a) AC waters at source, (b) DP waters at source, (c) AC waters in NBC, and (d) DP waters in NBC; relative number of particles from all 10 REF experiments per $0.1° \times 50$ m bin, i.e. local count divided by the total number of particles in the respective subset, in % (color shading); overlaid mean (2000-2009) Eulerian cross-section velocities in ms$^{-1}$ (light-grey contours, note the different contour intervals); and mean cumulative Lagrangian volume transport per 100 m depth bin in Sv (bar graphs).





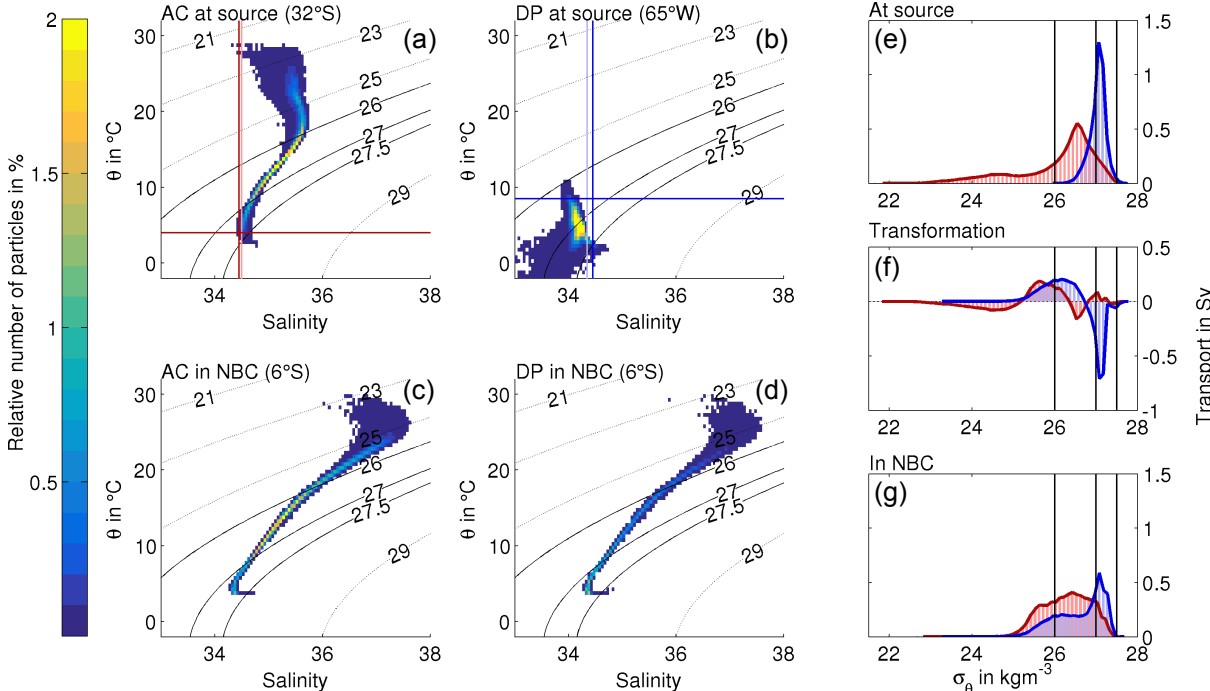

**Figure 7.** Mean potential temperature ($\theta$) and salinity (S) characteristics, as well as transformation in density space of waters with AC and DP origin. (a-d) Relative number of particles from all 10 REF experiments per 0.5 °C × 0.1 psu bin, i.e., local count divided by the total number of released particles in the respective subset, in %. (e-g) Mean volume transport per density class (in 0.1 kgm$^{-3}$ bins) at source regions and in NBC, as well as volumetric water mass transformation (red: waters of AC origin, blue: waters of DP origin). Potential density levels used to separate surface, central, intermediate (and deep) waters are highlighted by solid black lines; dark blue (dark red) lines mark T and S values that constitute the upper (lower) limit for 99 % of the DP (AC) water volumes, light blue (light red) lines the respective limit for 95 %.





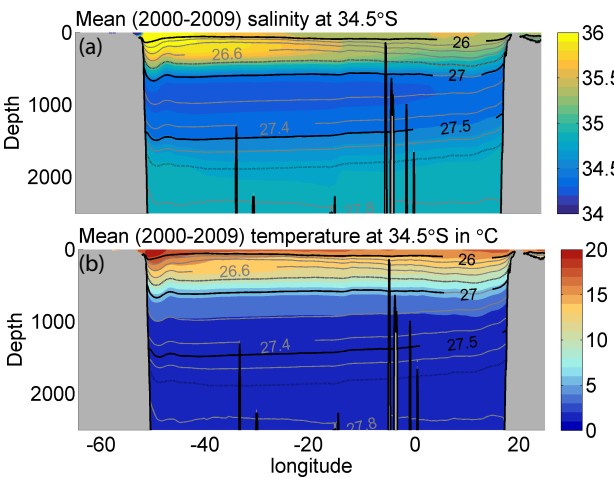

**Figure 8.** Mean (2000–2009) simulated Eulerian (a) salinity, and (b) potential temperature $\theta$ sections at 34.5°S (color shading); potential density anomalies are overlaid in grey contours — those used as separation of surface, central, intermediate, and deep waters are printed in black.





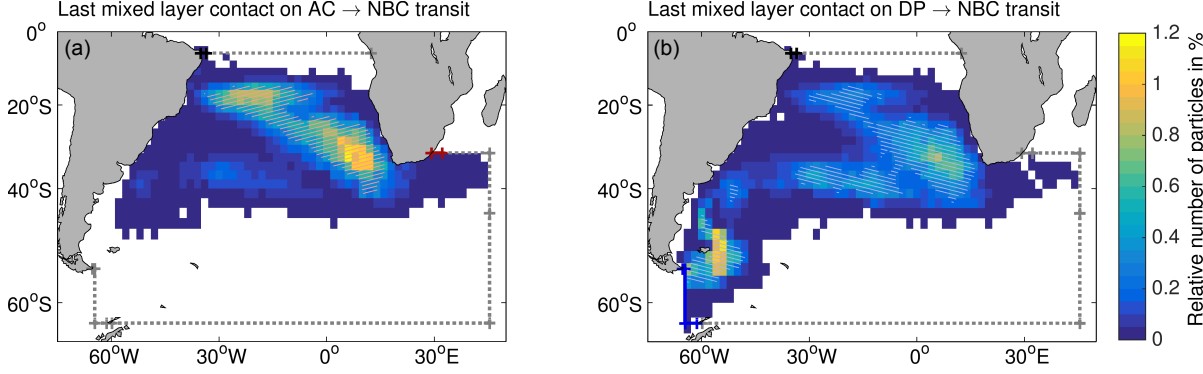

**Figure 9.** Horizontal distribution of the position of the last mixed layer contact for particles on the transit between (a) AC, or or (b) DP and NBC. Relative number of particles per $2° \times 2°$ bin, i.e. local count divided by the total number of particles from all 10 REF experiments in the respective subset that are entering the mixed layer at least once during their transit, in % (color shading); hatching highlights the most likely areas for the last mixed layer contact, encompassing 75 % of the respective set of particles entering the mixed layer.





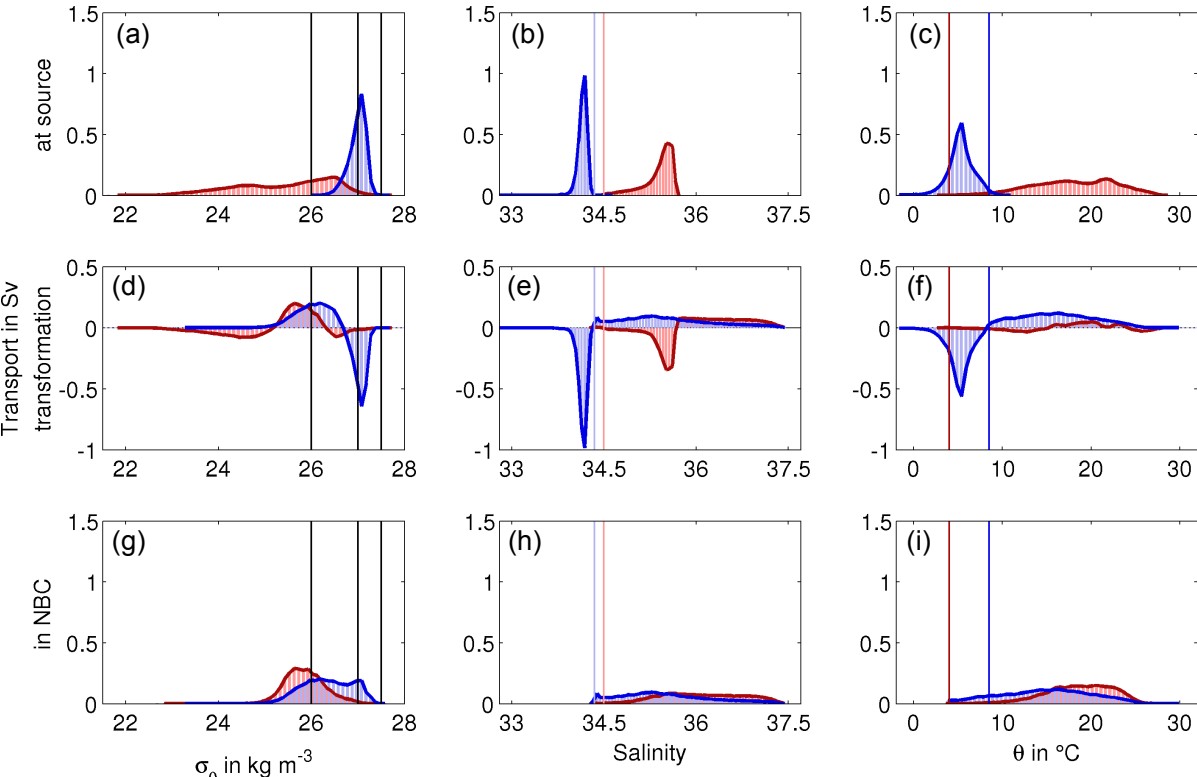

**Figure 10.** Mean property modification of waters with AC and DP origin that enter the mixed layer at least once during their transit. Mean volume transport per density (left, 0.1 kgm$^{-3}$ bins), salinity (middle, 0.05 psu bins), and temperature (right, 0.5 °C bins) class at source region (upper) and in NBC (lower), as well as volumetric property transformation (middle) for waters of AC (red bars) and DP (blue bars) origin. From all 10 REF experiments only particle trajectories with at least one mixed layer contact during their transit are considered. Potential density levels used to separate surface, central, intermediate and deep waters are highlighted by solid black lines; dark blue (red) lines mark θ and S values that constitute the upper (lower) limit for 99 % of the DP (AC) waters, light blue (red) lines represent the 95 % limit (cf. Fig. 7).





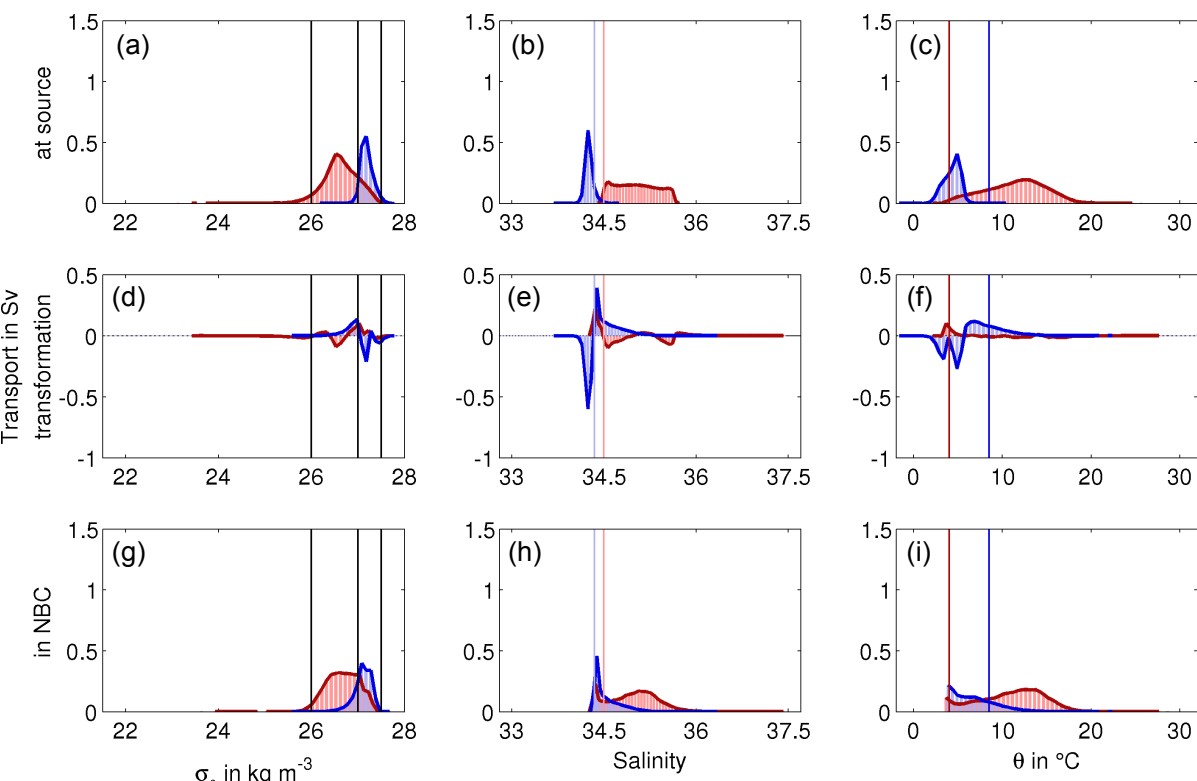

**Figure 11.** Same as Fig. 10 but for waters with AC and DP origin that that do not enter the mixed layer during their transit.



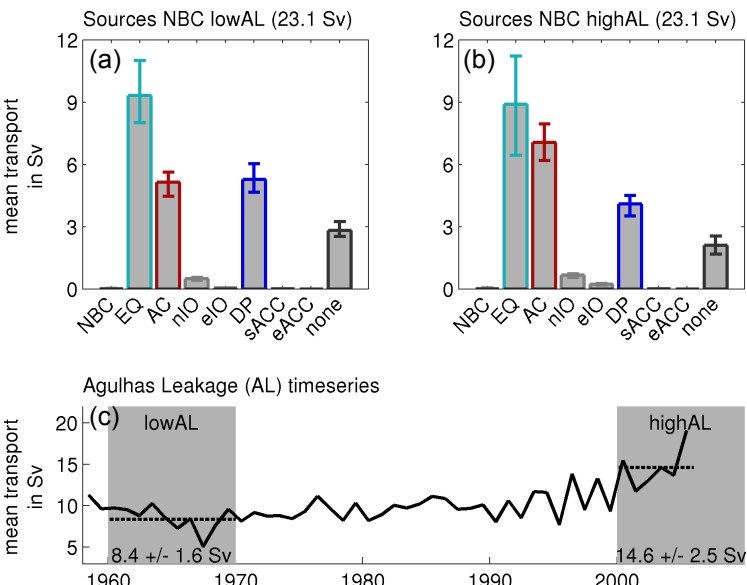

**Figure 12.** Sensitivity of NBC sources to the strength of Agulhas leakage (AL): mean volumetric contributions of the individual sources to the NBC inferred from (a) lowAL and (b) highAL (cf. Fig. 3); (c) interannual variability of AL transport as derived by Schwarzkopf et al. (in preparation, cf. Appendix A).



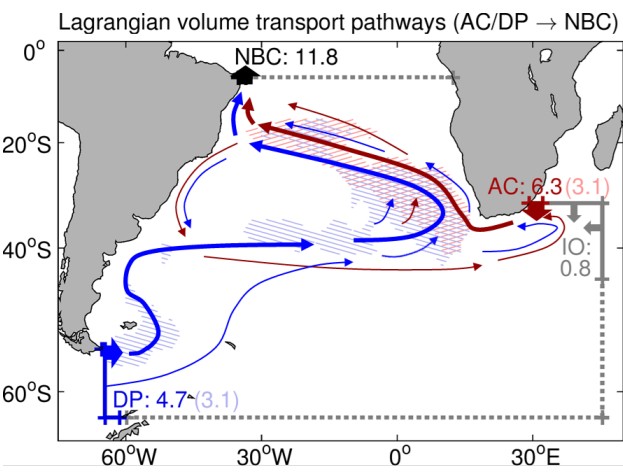

**Figure 13.** Summary of the AMOC's upper limb ‚cold' (DP, blue) and ‚warm' (mainly AC, red) water routes inferred from $O(10^6)$ simulated Lagrangian particle trajectories (REF experiments): major (thick arrows) and minor (thin arrows) advective pathways; Lagrangian mean cross-section transports in Sv (numbers); respective volumetric contribution of particles entering the mixed layer at least once during the transit (numbers in brackets); and most likely areas of last mixed layer contact (hatching, same as in Fig. 9).





| | | in AC | | | | in DP | | |
|---|---|---|---|---|---|---|---|---|
| | | surface | central | interm. | Σ | surface | central | interm. | Σ |
| | surface | (1.2) | 0.0 | 0.0 | 1.2 | (0.0) | 0.5 | 0.4 | 0.9 |
| in NBC | central | 0.8 | (3.4) | 0.2 | 4.4 | 0.0 | (1.0) | 1.3 | 2.3 |
| | interm. | 0.0 | 0.3 | (0.4) | 0.7 | 0.0 | 0.1 | (1.4) | 1.5 |
| | Σ | 2.0 | 3.7 | 0.6 | 6.3 | 0.0 | 1.6 | 3.1 | 4.7 |

**Table 1.** Net water mass transformation of surface, central, and intermediate waters with AC and DP origin between their entry into the South Atlantic and arrival in the NBC in Sv; water volumes without any net transformation into a different class are additionally listed in brackets.