# Peer review of "Cold vs. warm water route — sources for the upper limb of the AMOC revisited in a high-resolution ocean model"

_Ocean Science, 2018_

## Referee Comment (RC1) · Anonymous Referee #1 · 17 Jan 2019

I enjoyed reading this manuscript, which has a very clear narrative, is technically sound, and clearly answered the questions it set out to answer. The authors use well established Lagrangian methods and an eddy-rich model to update previous Lagrangian estimates of the South Atlantic AMOC return flow circulation patterns. Contrary to previous Lagrangian analyses based on coarser-resolution models, they find that the "cold-water" path through Drake Passage is a significant contribution to the AMOC upper-cell return flow (and how this might have changed over time). They also use the Lagrangian approach to clarify the water-mass transformation histories of the "warm/salty-water" and "cold/fresh-water" pathways as they interact with the mixed-layer. I have general comments about the framing of the paper and the lack of some important references. I

have only minor comments about the content of the methods and results.

General Comments:

The major weakness of the manuscript is its disregard for theories of the Atlantic Meridional Overturning Circulation and inter-basin exchange. The remainder of the general comments section is devoted to these theories.

In particular, I recommend an additional sentence before p.2, l.8-12 describing research about the salt advection feedback which maintains the AMOC and also the related (bi-) stability of the AMOC. The authors should modify the following lines (currently 8-12) to contextualize those references in this boarder theoretical context. It would be good to link back how the prevalence of the cold/fresh-water route fits into this context (perhaps not much, since as you have shown these waters are subsequently transformed as they traverse the South Atlantic).

Examples of conceptual theoretical models of the AMOC and the salt-advection feedback: Stommel's classic two-box model (https://onlinelibrary.wiley.com/doi/abs/10.1111/j.2153-3490.1961.tb00079.x) Rahmstorf's recent model (https://link.springer.com/article/10.1007/s003820050144) Wolfe and Cessi's more recent attempt (https://journals.ametsoc.org/doi/10.1175/JPO-D-13-0154.1)

Similarly, in p.2, l.26-33, the authors cite many observational inverse models and model-based Lagrangian analyses on both sides of the debate but there is very little theoretical consideration of why either the cold-water or warm-water pathways should be favored, despite the existence of such studies. Notably, Cessi and Jones (https://journals.ametsoc.org/doi/10.1175/JPO-D-16-0249.1) derive an elegant theoretical explanation for the (supposed) dominance of the warm/salty-water route relative to the cold/fresh-water route. Summarizing their theoretical argument, they say:

"Here, we present arguments indicating that all of the supergyre enables the warmroute
exchange of the MOC. Specifically, we show that the position of the short continent relative to the latitude of the zero Ekman pumping determines whether the cold route or the warm route is the preferred mode of interbasin exchange. We find that as long as there is a subtropical supergyre connecting the South Atlantic to the Indo-Pacific, the exchange is via the warm route. Conversely, if the subtropical gyres in the Atlantic and Indo-Pacific sectors are separated, then the interbasin flow is via the cold route."

Given that this theoretical result directly contradicts the findings of this paper, I would like the authors to acknowledge this work and perhaps speculate on what might be the cause of the contradiction.

Specific Comments:

p.5, l.8-9: Why free-slip in the base model and no-slip in the nest? Is this a standard difference between model formulations for eddy-permitting and eddy-rich models? I suspect it does not make much of a difference to the simulation but I am curious and other readers will likely be as well.

p.9, l.16-31: Perhaps worth mentioning that in this case (and more generally) it is difficult to directly compare Lagrangian studies because making slightly different choices in release / conditional sections results in differences in transport estimates for which it is impossible to disentangle model differences from methodological differences. The lack of Lagrangian analysis inter-comparisons makes it even more difficult to determine how important these differences can be. Some progress has been made by recent model inter-comparisons across OGCM and Lagrangian Model code by Tamsitt et al. (https://www.nature.com/articles/s41467-017-00197-0) and across model resolutions within a single OGCM and Lagrangian Model formulation by Drake et al. (https://agupubs.onlinelibrary.wiley.com/doi/full/10.1002/2017GL076045), both of which consider the complimentary pathways of upwelling circumpolar deep water in the Southern Ocean.

p.9, l.27-30: Please explain what a "OGCM used in diagnostic mode" means. Do you

mean in a data assimilation mode, as in the ECCO models?

p.10, l.6-9: It may be worth clarifying that the Lagrangian method used here does not account for the effects of sub-grid scale stirring, which would act to further disperse trajectories and could modify transit time distributions.

p.12, l.7: "alreday" is a typo for "already" but I would just leave the word out all together as the sentence does much make much sense with it in there.

p.12, l.10-11: Could you clarify whether these distributions (and all following distributions) are particle weighted and whether this is equivalent to area-weighting, transport-weighting, or neither? It seems that for T,S diagrams along a section that area-weighting might be most appropriate?

p.13, l.24-28: I am admittedly a bit confused about how the Ariane model considers flow in the mixed layer. Since transformations within the mixed layer are a key part of this section, I feel it would be important to discuss what it means to advect particles with the resolved velocities in a mixed layer for which (presumably) convection is not resolved (Is this true? I have little experience with such a high-resolution model). In most mixed layer schemes that I am familiar with, the tracer fields are rearranged or homogenized by artificially increasing the vertical diffusivity. What does this mean for a particle in the mixed layer? See for example the discussion in van Sebille et al. (https://agupubs.onlinelibrary.wiley.com/doi/full/10.1002/grl.50483).

p.17, l.11-13: See however Tamsitt et al (https://www.nature.com/articles/s41467-017-00197-0) and van Sebille et al (https://agupubs.onlinelibrary.wiley.com/doi/full/10.1002/grl.50483) who use particle probability maps to show that even in a Lagrangian framework in eddying models, the Global MOC still has some spatial coherence that corresponds to coherent tracer distributions.

Figure 7: It took me perhaps 10 minutes to understand what was happening in this entire figure. I would recommend adding the blue lines to (a) and the red lines to (b)

[Figure]

and perhaps adding annotations of "AC" in red and "DP" in the top left of (a). The use of parentheses to signify the converse in the last sentence of the caption did not help.

Technical Corrections:

p.5, l.13: Typo? Should the maximum bi-harmonic eddy viscosities Ahm0 be -6x10ˆ9 (-1.5x10ˆ11)?

––––––––––––––––––––––––––––

---

## Referee Comment (RC2) · Anonymous Referee #2 · 28 Jan 2019

In this study the authors outline very clearly a compelling Lagrangian analysis of the different sources of water to the NBC, and how their respective property changes are brought about though the South Atlantic. I recommend this study for publication in Ocean Science, though as described below, I think that improvements could be made to the manuscript with the inclusion of more discussion on the big picture implications of their findings.

Main comments:

Two of the big picture implications that I think would be particularly valuable to discuss are: 1) the potential implications of the different routes for Stommel's advective salt

feedback (or Fov; e.g. Drijfhout et al., 2011, Climate Dynamics). Model's often get this wrong, and the implications for this may be large (e.g. Liu et al., 2017, Science Advances). I wonder, therefore whether, the models could be getting the Fov sign wrong because they are underestimating the fresher DP contribution. 2) A comparison of the pathways to those produced in more idealized and theoretical studies, such as the recent papers by Spencer Jones and Paola Cessi. This would be useful since those simpler models are the ones we often rely on for clearer diagnoses of the mechanisms at play.

It would also be useful to have more description of the study by Rodrigues et al., (2010), the observations of which are used to validate this work. The authors outline in the introduction that the relative contributions from each source are strongly debated between many studies. Therefore, in order for the reader to accept this study as the most accurate among them, it will require that we agree the comparison to observations is better. However, I only found a three-line description of that observational study (P9L31-25).

Finally, it would help if the authors could provide more discussion of the perceived weaknesses of the experimental setup. While many of the earlier studies did not use high resolution models, this model has quite a short spin-up time and appears to only use interannual forcing fields. E.g. might higher resolution winds allow more water to cross from the AC?

Other comments:

- P12L21: This is an interesting argument, and the authors have convincingly demonstrated that the two water masses are made more distinct by to their salinity characteristics. However, what I think is probably more important in terms of how they should be labelled is the relative impacts the T and S differences have on density. While the water masses might be more easily delineated by salinity, it does not mean that those salinity differences have as big an impact on density as the temperature differences

(e.g. if the salinity range is smaller). Given the nonlinearity of the equation of state, it may not be trivial to fully estimate those impacts, but a rule-of-thumb estimation would still be useful. If it turns out that the temperature differences have a larger impact on density, then the warm- and cold-route terminology would likely remain preferable.

- P13L1-5: I am unclear on these density definitions. Wouldn't these density definitions of surface, central and intermediate waters depend on latitude, and therefore be different for the two sources of water? Some additional description may help.

- P14L14-17: This is an interesting result and a very nice analysis.

- P15L32: I don't understand this first sentence.

- P16L14: The wording of this sentence could do with some revision.

- P16L27: Here and elsewhere, "evoked" should be something more like 'induced'.

---

## Author Comment (AC2) · 18 Mar 2019

Many thanks for your positive feedback. We appreciate your suggestions and believe that including them strengthened the general framing and discussion of our work.

For detailed answers to the specific points you raised as well as respective changes made to the manuscript please refer to the supplementary pdf.

Please also note the supplement to this comment:
https://www.ocean-sci-discuss.net/os-2018-134/os-2018-134-AC2-supplement.pdf

---

## Author Response (AR2)

**Response to Anonymous Referee #1**

*Original reviewer's comments are inserted in black, Author Replies are added in blue, and Changes made to the Manuscript are finally listed in grey, whereby page and line numbers refer to the fully revised version of the manuscript.*

I enjoyed reading this manuscript, which has a very clear narrative, is technically sound, and clearly answered the questions it set out to answer. The authors use well established Lagrangian methods and an eddy-rich model to update previous Lagrangian estimates of the South Atlantic AMOC return flow circulation patterns. Contrary to previous Lagrangian analyses based on coarser-resolution models, they find that the "coldwater" path through Drake Passage is a significant contribution to the AMOC upper-cell return flow (and how this might have changed over time). They also use the Lagrangian approach to clarify the water-mass transformation histories of the "warm/salty-water" and "cold/fresh-water" pathways as they interact with the mixed-layer. I have general comments about the framing of the paper and the lack of some important references. I have only minor comments about the content of the methods and results.

AR: *Many thanks for this encouraging reply and constructive criticism. We are happy that you enjoyed reading our manuscript and that you generally agree to the employed methodology as well as to the presentation and interpretation of the results. We greatly appreciate your suggestions regarding the framing of the paper and are convinced that they helped to improve the manuscript.*

**General Comments:**

The major weakness of the manuscript is its disregard for theories of the Atlantic Meridional Overturning Circulation and inter-basin exchange. The remainder of the general comments section is devoted to these theories.

AR: *Thank you for pointing this out and providing such detailed suggestions for improvements.*

▪ In particular, I recommend an additional sentence before p.2, l.8-12 describing research about the salt advection feedback which maintains the AMOC and also the related (bi-) stability of the AMOC. The authors should modify the following lines (currently 8-12) to contextualize those references in this boarder theoretical context. It would be good to link back how the prevalence of the cold/fresh-water route fits into this context (perhaps not much, since as you have shown these waters are subsequently transformed as they traverse the South Atlantic). Examples of conceptual theoretical models of the AMOC and the salt-advection feedback: Stommel's classic two-box model (https://onlinelibrary.wiley.com/doi/abs/10.1111/j.2153-3490.1961.tb00079.x) Rahmstorf's recent model (https://link.springer.com/article/10.1007/s003820050144) Wolfe and Cessi's more recent attempt (https://journals.ametsoc.org/doi/10.1175/JPO-D-13-0154.1)

AR: *We agree that the theories regarding the salt advection feedback are closely related to the topic of our manuscript and help supporting its relevance, hence we gladly incorporated your suggestions. However, as clearly stated, e.g., in the abstract, results, and summary, our study does not claim a prevalence of the cold water route. We argue that even though the warm water route is the dominant contributor to the AMOC's upper limb, the cold water route contribution is with around 40% not negligible. To address the question how these results fit into the context of the salt advection feedback, we followed Drijfhout et al. (2011) and calculated the AMOC related freshwater transport (hereafter Fov) across the southern boundary of the Atlantic. At 30°S, Fov amounts to -124.52. The negative sign implies a net AMOC-driven southward transport of freshwater out of the Atlantic. This suggests that the freshwater input through our relative high DP contribution is still dominated by the salt input through the AC. Following the theories of the salt advection feedback, the negative Fov has a strengthening effect on the AMOC, but also enables a positive feedback with destabilizing effect on the AMOC theoretically allowing for the*

*'off' state. Hence, the existence of the cold/fresh water route in our model simulation does not disagree with the theory of a potentially bi-stable state of the AMOC related to a negative Fov. Note though that the reliability of Fov as an AMOC stability criterion is currently debated (cf., Gent 2018; Cheng et al., 2018) and further studies are needed to unravel the potential impact of changes in the AC and DP water contributions not only on the strength, but also on the stability of the AMOC.*

*CM: We now introduce the salt advection feedback in the introduction and relate our results to the respective theories in the conclusions by making use of the estimated freshwater transport across the southern boundary of the Atlantic.*

*p.2, ll.13-20: "In particular, it has been suggested that a net southward freshwater transport related to the northward advection of relatively high saline waters along the AMOC's upper limb in the South Atlantic introduces a positive feedback (e.g., Stommel, 1961; Rahmstorf, 1996; Drijfhout et al., 2011): a weakening (strengthening) of the meridional overturning circulation in the Atlantic (AMOC) results in reduced (enhanced) northward salt transport and corresponding freshening (salinification) of the North Atlantic, and further weakening (strengthening) of the overturning. This salt-advection feedback constitutes the basis for the theory of rapid climate shifts associated with a bi-stability of the AMOC with either vigorous overturning ('on' state) or weak/reversed overturning ('off' state) (e.g., Rahmstorf, 2002; Deshayes et al., 2013)."*

*p.18, ll.6-21: Considering the relatively high contribution of fresh DP waters to the upper limb transport of the AMOC revealed by our study, it arises the question how this fits into the context of the salt advection feedback. In the employed hindcast simulation, the average (1958-2009) AMOC related freshwater transport (hereafter Fov, calculated following Drijfhout et al. (2011)) across the southern boundary of the Atlantic at 30 °S amounts to -124.52 mSv, which is in line with estimates for other hindcast simulations with OGCM configurations at comparable resolution (c.f. Deshayes et al., 2013). The negative sign implies a net AMOC related southward transport of freshwater out of the Atlantic or, equivalently, a northward advection of salt into the Atlantic. This suggests that the freshwater input through the DP contribution is still dominated by the salt input through the AC contribution. Following the theories of the salt advection feedback, the negative Fov has a strengthening effect on the current AMOC, but also enables a positive destabilizing feedback theoretically allowing for the 'off' state. Hence, the relative large DP in our model simulation does not disagree with the theory of a potentially bi-stable state of the AMOC related to a negative Fov. Note though that the reliability of Fov as an AMOC stability criterionis currently debated (cf. Gent, 2018; Cheng, 2018). The results of our Lagrangian analysis also put a fundamental assumption of the salt-advection feedback into question, namely, that anomalies in the freshwater transport at the southern boundary of the Atlantic are coherently advected into the North Atlantic. The substantial along-track property modifications revealed in this study challenge the use of the inferred advective volume transport pathways and timescales for assessing the pathways and timescales with that upper ocean temperature or salinity anomalies are transmitted through the Atlantic."*

- Similarly, in p.2, l.26-33, the authors cite many observational inverse models and model-based Lagrangian analyses on both sides of the debate but there is very little theoretical consideration of why either the cold-water or warm-water pathways should be favored, despite the existence of such studies. Notably, Cessi and Jones (https://journals.ametsoc.org/doi/10.1175/JPO-D-16-0249.1) derive an elegant theoretical explanation for the (supposed) dominance of the warm/salty-water route relative to the cold/fresh-water route. Summarizing their theoretical argument, they say: "Here, we present arguments indicating that all of the supergyre enables the warmroute exchange of the MOC. Specifically, we show that the position of the short continent relative to the latitude of the zero Ekman pumping determines whether the cold route or the warm route is the preferred mode of interbasin exchange. We find that as long as there is a subtropical supergyre connecting the South Atlantic to the Indo-Pacific, the exchange is via the warm route. Conversely, if the subtropical gyres in the Atlantic and Indo-Pacific sectors are separated, then the interbasin flow is via the cold route." Given that this theoretical result directly

contradicts the findings of this paper, I would like the authors to acknowledge this work and perhaps speculate on what might be the cause of the contradiction.

*AR: We agree that the theoretical considerations by Cessi and Jones (2017) need to be included in the introduction for a thorough and complete review of the existing literature. However, we do not think that their conclusion based on an idealized coarse-resolution model set-up, namely, that the upper limb's exchange occurs exclusively via the warm water route, contradict our study, since (i) we do not claim a prevalence of the cold water route, but still support the dominance of the warm water route (see comments above), and (ii) Cessi and Jones (2017) state themselves that under more realistic model set-ups mixed exchange routes may exist.*

*CM: We are now taken into account the study by Cessi and Jones (2017) by introducing the following paragraphs to the introduction:*

*p.3, ll.20-27: "Recently, [the warm water route hypothesis] got further theoretical support by Cessi and Jones (2017), who studied the upper limb of the AMOC in an idealized model configuration with simplified atmospheric forcing and geometry (one wide and one narrow basin, representing the Indo-Pacific and Atlantic Oceans, separated by a long and a short continent, representing America and Eurasia/Africa, and connected in the South through a reentrant channel, representing the ACC). They showed that the latitude of zero Ekman pumping relative to the southern extent of the short continent determines the route of the upper limb's interbasin exchange; and that under the current geographical settings, which allow for a southern hemisphere supergyre, the exchange occurs exclusively south of the short continent, that is, via the warm water route."*

*p.3, ll.28-34: "Still, the studies supporting the warm water route hypothesis remain inconclusive. On the one hand, the idealized model configuration of Cessi and Jones (2017) leads to an artificial separation of warm and cold water route scenarios, whereby realistic wind stress forcing and geometry indeed may allow for mixed exchange routes as stated by the authors themselves. On the other hand, Cessi and Jones (2017) and most other studies in support of the warm water route hypothesis were based on the evaluation of relatively coarse resolution non-eddying or eddy-permitting ocean model simulations (Speich et al., 2001; Donners and Drijfhout, 2004; Speich et al., 2007), and various studies (e.g., Biastoch et al., 2008c; Durgadoo et al., 2013) have demonstrated that coarse non-eddying ocean models overestimate the strength of Agulhas leakage."*

**Specific Comments:**

- p.5, l.8-9: Why free-slip in the base model and no-slip in the nest? Is this a standard difference between model formulations for eddy-permitting and eddy-rich models? I suspect it does not make much of a difference to the simulation but I am curious and other readers will likely be as well.

  *AR: Free-slip has been standard for ORCA025 model configurations at ¼° resolution (Barnier et al., 2006) so we adopted that boundary condition also for our ORCA025 base. Sensitivity experiments with the employed eddy-rich model configuration performed by Schwarzkopf et al. (2019), show that at this higher resolution no-slip yields better performance regarding the circulation features of interest in this study.*

  *CM: To avoid confusion, we deleted the information on the lateral boundary conditions from our manuscript and instead refer for details on the experimental set-up to the companion paper of Schwarzkopf et al. (2019), who thoroughly introduce the employed model configuration, including a discussion of the sensitivity of the simulations with respect to the choice of lateral boundary conditions.*

  *p.6, l.2: "For more details on the experimental set-up and a general model validation please refer to Schwarzkopf et al. (2019)."*

- p.9, l.16-31: Perhaps worth mentioning that in this case (and more generally) it is difficult to directly compare Lagrangian studies because making slightly different choices in release /

conditional sections results in differences in transport estimates for which it is impossible to disentangle model differences from methodological differences. The lack of Lagrangian analysis inter-comparisons makes it even more difficult to determine how important these differences can be. Some progress has been made by recent model inter-comparisons across OGCM and Lagrangian Model code by Tamsitt et al. (https://www.nature.com/articles/s41467-017-00197-0) and across model resolutions within a single OGCM and Lagrangian Model formulation by Drake et al. (https://agupubs.onlinelibrary.wiley.com/doi/full/10.1002/2017GL076045), both of which consider the complimentary pathways of upwelling circumpolar deep water in the Southern Ocean.

*AR: We agree that, generally, a direct comparison of Lagrangian studies can be very difficult due to potential sensitivities of the results to the chosen release and sampling sections, integration and interpolation schemes, and details in the integration strategy such as the time step, number of released particles, or boundary conditions, as nicely discussed in Tamsitt et al. (2018). However, the Lagrangian model studies we are referring to for the comparison of the transport estimates all applied the same linear interpolation as well as analytical integration method to volume-conserving flow fields. Moreover, the analytical integration method works without any time stepping and respects the volume conservation of the underlying flow field thereby avoiding coast crashes (for details see, e.g., van Sebille et al., 2018). Hence, we argue, that the sensitivities discussed in Tamsitt et al. (2018) are of minor importance for this study and the cited transports indeed mostly differ due to differences in the underlying OGCM output as well as due to differences in the release and sampling sections, most importantly, the reference section for the Lagrangian decomposition of the AMOC.*

*CM: To better account for this matter we added the following:*

*p.10, ll.11-14: "[…] All these studies employed the same analytical trajectory integration method so that differences in the derived volumetric contributions to the upper limb of the AMOC can be mainly related to differences in the analyzed OGCM output (even though the different reference sections for the Lagrangian decomposition do not allow for a detailed one-to-one comparison)."*

- p.9, l.27-30: Please explain what a "OGCM used in diagnostic mode" means. Do you mean in a data assimilation mode, as in the ECCO models?
  *AR: "Robust diagnostic mode" is a commonly used expression in ocean modelling. It stands for the procedure in which during run-time the models tracer fields are frequently relaxed towards climatology (e.g. Levitus) to reduce spurious model drifts.*

- p.10, l.6-9: It may be worth clarifying that the Lagrangian method used here does not account for the effects of sub-grid scale stirring, which would act to further disperse trajectories and could modify transit time distributions.
  *CM: We clarify this by adding respective sentences to the method section and to the paragraph on transit times*
  *p.7, l.9: "No additional sub-grid scale Lagrangian diffusion parametrization was implemented."*
  *p.11, ll.4-6: "Here we infer advective timescales from the simulated volume transport trajectories. Note that timescales inferred from trajectories accounting for the effect of sub-grid scale physics, e.g., representing advective-diffusive tracer spreading, could lead to modified, e.g., broadened, transit time distributions."*

- p.12, l.7: "alreday" is a typo for "already" but I would just leave the word out all together as the sentence does much make much sense with it in there.
  *CM: We deleted it*

- p.12, l.10-11: Could you clarify whether these distributions (and all following distributions) are particle weighted and whether this is equivalent to area-weighting, transport-weighting, or neither? It seems that for T,S diagrams along a section that area-weighting might be most appropriate?

*AR: In fact, in the original manuscript, all plots showing color-shaded distributions were based on simple particle frequencies per bin (as, to our mind, clearly stated in the figure captions as well as in the text). This should have allowed for a first qualitative assessment by means of a measure that is relatively easy to understand. For all further quantitative assessments, we referred to the bar graphs of transport-weighted distributions. Even though simple particle frequency distributions theoretically differ from transport-weighted distributions (since each particle can be associated with a slightly different transport) and area-weighted distributions (since, e.g., lon/lat bins do not necessarily have the same size), these differences are negligible for the qualitative assessment of the integrated measures of interest in this study.*

*CM: Nevertheless, to avoid confusions, we now changed simple particle frequencies to transport-weighted particle frequencies in Figures 6, 7, 9 (and Figure 13) and the figure captions have been adjusted accordingly. Moreover, a short explanation has been added to the method section:*

*p.8, ll.8-11: "To visualize the distribution of AC or DP waters along distinct sections we inferred binned transport-weighted particle frequencies by dividing the cumulative transport of particles occupying a certain bin by the cumulative transport of the whole set of particles. Transport-weighted particle distributions are preferred over simple particle frequency distributions since they take into account that each particle can be associated with a slightly different transport."*

- p.13, l.24-28: I am admittedly a bit confused about how the Ariane model considers flow in the mixed layer. Since transformations within the mixed layer are a key part of this section, I feel it would be important to discuss what it means to advect particles with the resolved velocities in a mixed layer for which (presumably) convection is not resolved (Is this true? I have little experience with such a high-resolution model). In most mixed layer schemes that I am familiar with, the tracer fields are rearranged or homogenized by artificially increasing the vertical diffusivity. What does this mean for a particle in the mixed layer? See for example the discussion in van Sebille et al. (https://agupubs.onlinelibrary.wiley.com/doi/full/10.1002/grl.50483).

  *AR: ARIANE has been developed as a Lagrangian analysis tool, that means, it purely advects particles with the simulated volume-conserving flow field of an OGCM along analytically integrated streamlines – without any Lagrangian sub-grid scale parametrizations. Hence, the resulting particle trajectories represent volume transport pathways fully determined by the resolved flow. OGCM sub-grid scale parametrizations – such as tracer diffusion and, more specifically, parametrizations for vertical tracer mixing in the mixed layer (which is achieved, as you described above, by increasing the vertical diffusivities) – are "only" implicitly included through tracer changes along the particle trajectories.*

  *In contrast, tools as CMS – which has been employed in the above mentioned example of van Sebille et al. – allow for Lagrangian modelling attempts to directly simulate tracer spreading (instead of volume transport pathways). They add Lagrangian parametrizations to the particle advection to explicitly account for sub-grid scale physics. Consequently, mixing should be represented by a redistribution of particles and the tracer values for one particle along its trajectory should stay/ be assumed constant with time. However, it is still debatable whether the applied relatively simplistic parametrizations indeed result in an adequate representation of tracer spreading.*

  *CM: We adjusted the description of the particle-tracking method to further clarify the treatment of sub-grid scale parametrizations.*

  *p.7, ll.7-12: "ARIANE is a freely available FORTRAN software that infers Lagrangian particle trajectories from simulated three-dimensional volume-conserving velocity fields saved on a C-grid by offline advecting virtual fluid particles along analytically computed streamlines. No additional sub-grid scale Lagrangian diffusion parametrization was implemented. The obtained trajectories thus represent volume transport pathways, which may experience along-track tracer and density changes that are reflecting the sub-grid scale parametrizations of the underlying OGCM, including vertical tracer mixing in the mixed layer. For a detailed discussion of this concept please refer to van Sebille et al. (2018)."*

- p.17, l.11-13: See however Tamsitt et al (https://www.nature.com/articles/s41467-017-00197-0) and van Sebille et al (https://agupubs.onlinelibrary.wiley.com/doi/full/10.1002/grl.50483) who use particle probability maps to show that even in a Lagrangian framework in eddying models, the Global MOC still has some spatial coherence that corresponds to coherent tracer distributions.

  *AR: We did not want to give the impression that there is no spatial coherence at all, but that the spatial coherence is limited. In fact, in the paragraph below the one you are referring to, we describe that some spatial coherence exists also in our simulations on the intermediate water level.*

  *CM: To further clarify this point we now explicitly mention the remaining spatial coherence of the AMOC:*

  *p.18, ll.25-27: "It is noteworthy though that the deeper parts of the upper limb, such as the intermediate waters transiting the South Atlantic without mixed layer contact, largely keep their characteristic properties along their transit, indicating some remaining spatial coherence of the AMOC."*

- Figure 7: It took me perhaps 10 minutes to understand what was happening in this entire figure. I would recommend adding the blue lines to (a) and the red lines to (b) and perhaps adding annotations of "AC" in red and "DP" in the top left of (a). The use of parentheses to signify the converse in the last sentence of the caption did not help.

  *AR: We are sorry that this figure appeared so complicated.*

  *CM: We added the lines as suggested, and completely rewrote the figure caption:*

  *p.33: "Thermohaline properties of waters with AC and DP origin inferred from all 10 REF experiments. (a-d) Mean potential temperature (θ) and salinity (S) characteristics of AC and DP waters at their source and within the NBC: relative transport-weighted particle frequency per 0.5 °C x 0.1 psu bin in % (color shading); initially, 99 % of the DP (AC) water volume can be found at temperatures colder (warmer) than 8.5 °C (4.0 °C) and at salinities lower (greater) than 34.45, as indicated by blue (red) lines. (e-g) Mean volume transport per density class (in 0.1 kgm$^3$ bins) of AC (red) and DP (blue) waters at their source and within the NBC, as well as associated transformation in density space (bar graphs); potential density levels used to separate upper, intermediate, and deep waters are highlighted by solid black lines."*

**Technical Corrections:**

- p.5, l.13: Typo? Should the maximum bi-harmonic eddy viscosities Ahm0 be -6x10^9 (-1.5x10^11)?

  *CM: Thanks for spotting this typo, we corrected it.*

**Response to Anonymous Referee #2**

*Original reviewer's comments are inserted in black, Author Replies are added in blue, and Changes made to the Manuscript are finally listed in grey, whereby page and line numbers refer to the fully revised version of the manuscript.*

In this study, the authors outline very clearly a compelling Lagrangian analysis of the different sources of water to the NBC, and how their respective property changes are brought about though the South Atlantic. I recommend this study for publication in Ocean Science, though as described below, I think that improvements could be made to the manuscript with the inclusion of more discussion on the big picture implications of their findings.

*AR: Many thanks for your positive feedback. We appreciate your suggestions and believe that including them strengthened the general framing and discussion of our work.*

**Main comments:**

- Two of the big picture implications that I think would be particularly valuable to discuss are:
  1) the potential implications of the different routes for Stommel's advective salt feedback (or Fov; e.g. Drijfhout et al., 2011, Climate Dynamics). Model's often get this wrong, and the implications for this may be large (e.g. Liu et al., 2017, Science Advances). I wonder, therefore whether, the models could be getting the Fov sign wrong because they are underestimating the fresher DP contribution.

     *AR: We now added one paragraph to the introduction and one paragraph to the conclusions to relate our work to the theories to the salt advection feedback (**please refer to the response to referee#1 for details**). However, we do not think that we can justify a general statement on the relation between the AC/DP partitioning and deficiencies with respect to model's representations of AMOC stability, since (i) most state-of-the art OGCMs have a negative Fov, and also many CMIP5 climate models seem to get the sign right (even though Liu et al., (2017), stated otherwise, cf. Gent (2018)), and (ii) it is currently debated whether Fov is a reliable stability criterion at all (cf. Gent (2018), Cheng (2018)).*

  2) A comparison of the pathways to those produced in more idealized and theoretical studies, such as the recent papers by Spencer Jones and Paola Cessi. This would be useful since those simpler models are the ones we often rely on for clearer diagnoses of the mechanisms at play.

     *AR: We agree that the theoretical considerations by Cessi and Jones (2017) need to be included in the introduction for a thorough and complete review of the existing literature and added respective paragraphs to the manuscript (**please refer to the response to referee#1 for details**).*

- It would also be useful to have more description of the study by Rodrigues et al., (2010), the observations of which are used to validate this work. The authors outline in the introduction that the relative contributions from each source are strongly debated between many studies. Therefore, in order for the reader to accept this study as the most accurate among them, it will require that we agree the comparison to observations is better. However, I only found a three-line description of that observational study (P9L31-25).

     *AR: We understand that the details of the study by Rodrigues et al (2010) may be of interest to the reader, but we are of the opinion that our comparison that focuses on the AC and DP contributions is adequate for the purpose of the manuscript. On the one hand, we give several complementary reasons, why a solution with a non-negligible DP contribution may be the most realistic one (please also see the changes made to the introduction). In fact, to our mind, a good agreement with Rodrigues et al. (2010) alone is not sufficient to accept our study as the most accurate, given the limited spatial and temporal resolution of observational data. On the other hand, a more detailed*

*comparison between Rodrigues et al. 2010 and our results in terms of other derived quantities would require complex analysis which are beyond the scope and framing of this study and may be unnecessary, given the detailed model validation performed by us (see method section) and Schwarzkopf et al. (2019).*

- Finally, it would help if the authors could provide more discussion of the perceived weaknesses of the experimental setup. While many of the earlier studies did not use high resolution models, this model has quite a short spin-up time and appears to only use interannual forcing fields. E.g. might higher resolution winds allow more water to cross from the AC?

  *AR: It seems our formulations regarding the temporal resolution of the forcing fields has been misleading. The term interannual forcing has been used to contrast the forcing from the employed hindcast spanning the period 1958-2009 to that from a climatological run with no interannual forcing variability. More precisely, the employed atmospheric forcing for the period 1958-2009 from the CORE data set includes 6-hourly atmospheric state variables at 10m height (temperature, humidity and horizontal wind components), daily long and short-wave radiation (prior to 1984 based on a climatological mean annual cycle), and monthly precipitation (prior to 1979 based on a climatological mean annual cycle) as described in the listed reference.*

  *Even though the employed spin-up is clearly too short for the deep ocean to reach a stable state, it is rather long for a realistic ocean model configuration at such high resolution. It is fair to assume that at least the upper ocean, which is most relevant for this study, has reached an adequately adjusted state.*

  *CM: We reformulated all parts referring to interannually varying atmospheric forcing fields.*

  *p.5, ll.24-25: "(…) and subsequently run with forcing from the atmospheric fields of the Coordinated Ocean-Ice Reference Experiments data set version 2 (CORE; Large and Yeager, 2009; Griffies et al., 2009) for the period 1958–2009."*

  *p.11, ll.29-31: "(…), we used 5-day mean velocity fields of a hindcast experiment, whereas Speich et al. (2001) used monthly means from a climatological experiment. The increase in resolution and allowance for interannual variability most likely lead to (…)"*

**Other comments:**

- P12L21: This is an interesting argument, and the authors have convincingly demonstrated that the two water masses are made more distinct by to their salinity characteristics. However, what I think is probably more important in terms of how they should be labelled is the relative impacts the T and S differences have on density. While the water masses might be more easily delineated by salinity, it does not mean that those salinity differences have as big an impact on density as the temperature differences (e.g. if the salinity range is smaller). Given the nonlinearity of the equation of state, it may not be trivial to fully estimate those impacts, but a rule-of-thumb estimation would still be useful. If it turns out that the temperature differences have a larger impact on density, then the warm- and cold-route terminology would likely remain preferable.

  *AR: We agree that this a very interesting point worth of further dedicated analysis, which are, however, beyond the scope of the current study. Given that the positive temperature anomalies introduced by the inflow of upper waters South of Africa have been suggested to be dampened way faster than the positive salinity anomalies (e.g., Gordon, 1992) we anticipate a larger impact of the salinity difference. The importance of the salinity difference is further supported by the newly included discussion of the salt advection feedback (see comment above). However, overall, the relative importance of the temperature and salinity differences may be dependent on the details of the research question.*

  *Gordon, A. L., Weiss, R. F., Smethie, W. M., & Warner, M. J. (1992). Thermocline and intermediate water communication between the south Atlantic and Indian oceans. Journal of Geophysical Research, 97(C5), 7223. https://doi.org/10.1029/92JC00485*

*CM: The respective paragraph has been adjusted:*
*p.13, ll.19-22: "Hence, we may consider fresh and salty routes as an alternative and more precise terminology, which also accounts for the relative role of the two sources with respect to the salt advection feedback. Yet, dependent on the specific research question, the mean temperature difference between the two may still be of (larger) importance. Therefore, we would recommend referring directly to the geographic origin to avoid ambiguities."*

- P13L1-5: I am unclear on these density definitions. Wouldn't these density definitions of surface, central and intermediate waters depend on latitude, and therefore be different for the two sources of water? Some additional description may help.
  *AR: The density criteria for separating surface, central, and intermediate waters are – as we do acknowledge in lines 1-6 at page 14 – not uniquely defined. However, the chosen (or very similar) values have been meaningful applied for broader scale analysis in the subtropical and tropical South Atlantic (see references within the manuscript itself). In particular, Antarctic Intermediate Water (AAIW) can be detected in the given density range within the whole South Atlantic basin north of the Subantarctic Front, encompassing the upper limb pathways of the AC as well as DP contribution (cf. section 3.3. of main manuscript and, e.g., Table 2 of Heywood and King, 2002). Everything above the AAIW layer constitutes the upper water layer, which in the subtropical and tropical Atlantic can be further divided in central waters remotely formed by subduction and surface waters under direct influence of local air-sea fluxes. The density level that separates these two layers is indeed latitude dependent, given that the central water range broadens towards the tropics where it includes more and more varieties of subducted surface waters from the subtropics. Hence, we agree that the discussion of surface and central water transformation in the current form may be confusing.*
  *Heywood, K. J., & King, B. A. (2002). Water masses and baroclinic transports in the South Atlantic and Southern oceans. Journal of Marine Research, 60(5), 639–676. https://doi.org/10.1357/002224002762688687*
  *CM: We decided to no longer differentiate central and surface waters but instead combine the two into one category termed 'upper waters', which can be clearly separated from intermediate and deep waters by the chosen density criteria within the whole area of interest. The table, as well as Figures 7,8,10 and 11 and corresponding figure captions have been adjusted accordingly and the respective parts in the results section have been re-written.*

- P14L14-17: This is an interesting result and a very nice analysis.
  *AR: Thanks*

- P15L32: I don't understand this first sentence.
  *AR: We agree that this first sentence may be confusing if one assumes that the terms 'Agulhas leakage' and 'AC contribution to the AMOC' can be used interchangeable. However, in our study, the AC contribution to the AMOC always refers to the contribution of waters with Agulhas origin that make it into the tropics and become part of the NBC. Hence, the question rather is whether we can detect the increase in Agulhas Leakage as an increased contribution of Agulhas waters to the AMOC's upper limb further downstream in the tropics.*
  *CM: We now specified that we are referring to the AMOC contribution in the tropics in this particular sentence as well as elsewhere in the manuscript:*
  *p.16, ll.30-31: "The increase in the AC contribution to the AMOC's upper limb in the tropics is, however, not directly proportional to the increase in Agulhas Leakage, but weaker (1.9 Sv compared to 6.2 Sv, respectively)."*

- P16L14: The wording of this sentence could do with some revision.
  *CM: We split the sentence into two:*
  *p.17, ll.11-14: "To do so, we performed Lagrangian analyzes using 5-day mean output from a hindcast experiment (1958–2009) with the high-resolution (1/20°) ocean general circulation model INALT20. We employed the Lagrangian tool ARIANE to calculate O(10^6) advective volume*

*transport trajectories as well as along-track thermohaline property changes between the two source regions and the North Brazil Current (NBC), which channels the upper limb flow in the tropics."*

- P16L27: Here and elsewhere, "evoked" should be something more like 'induced'
  *CM: We exchanged 'evoked' by induced' within the whole manuscript*

[revised manuscript text omitted]